# Handover Management for Drones in Future Mobile Networks—A Survey

**DOI:** 10.3390/s22176424

**Published:** 2022-08-25

**Authors:** Ibraheem Shayea, Pabiola Dushi, Mohammed Banafaa, Rozeha A. Rashid, Sawsan Ali, Mohd Adib Sarijari, Yousef Ibrahim Daradkeh, Hafizal Mohamad

**Affiliations:** 1Department of Electronics and Communication Engineering, Istanbul Technical University (ITU), 34467 Istanbul, Turkey; 2Wireless Communication Centre, School of Electrical Engineering, Faculty of Engineering, Universiti Teknologi Malaysia, UTM, Johor Bahru 81310, Johor, Malaysia; 3Telecommunication Software and System Research Group, Communication Engineering Department, School of Electrical Engineering, Faculty of Engineering, Universiti Teknologi Malaysia, UTM, Skudai 81310, Johor, Malaysia; 4Department of Computer Engineering, University of Ha’il, Ha’il 55211, Saudi Arabia; 5Department of Computer Engineering and Networks, College of Engineering in Wadi Alddawasir, Prince Sattam bin Abdulaziz University, Al-Kharj 16436, Saudi Arabia; 6Faculty of Engineering and Built Environment, Universiti Sains Islam Malaysia, Bandar Baru Nilai, Nilai 71800, Negeri Sembilan, Malaysia

**Keywords:** drone, drone network, connected drone, Unmanned Aerial Vehicle (UAV), handover decision algorithm, handover management, mobility management, Fifth Generation (5G), Sixth Generation (6G), mobile networks, mobile ad hoc networks

## Abstract

Drones have attracted extensive attention for their environmental, civil, and military applications. Because of their low cost and flexibility in deployment, drones with communication capabilities are expected to play key important roles in Fifth Generation (5G), Sixth Generation (6G) mobile networks, and beyond. 6G and 5G are intended to be a full-coverage network capable of providing ubiquitous connections for space, air, ground, and underwater applications. Drones can provide airborne communication in a variety of cases, including as Aerial Base Stations (ABSs) for ground users, relays to link isolated nodes, and mobile users in wireless networks. However, variables such as the drone’s free-space propagation behavior at high altitudes and its exposure to antenna sidelobes can contribute to radio environment alterations. These differences may render existing mobility models and techniques as inefficient for connected drone applications. Therefore, drone connections may experience significant issues due to limited power, packet loss, high network congestion, and/or high movement speeds. More issues, such as frequent handovers, may emerge due to erroneous transmissions from limited coverage areas in drone networks. Therefore, the deployments of drones in future mobile networks, including 5G and 6G networks, will face a critical technical issue related to mobility and handover processes due to the main differences in drones’ characterizations. Therefore, drone networks require more efficient mobility and handover techniques to continuously maintain stable and reliable connection. More advanced mobility techniques and system reconfiguration are essential, in addition to an alternative framework to handle data transmission. This paper reviews numerous studies on handover management for connected drones in mobile communication networks. The work contributes to providing a more focused review of drone networks, mobility management for drones, and related works in the literature. The main challenges facing the implementation of connected drones are highlighted, especially those related to mobility management, in more detail. The analysis and discussion of this study indicates that, by adopting intelligent handover schemes that utilizing machine learning, deep learning, and automatic robust processes, the handover problems and related issues can be reduced significantly as compared to traditional techniques.

## 1. Introduction

The drone, also known as an Unmanned Aerial Vehicle (UAV), is an autonomously flying aircraft controlled by an individual. *In this paper, the terms drone and UAV are used interchangeably*. Drones offer benefits such as low-cost access, effortless data collection, high efficiency, fewer hazards to humans, and logistical support. Based on their potential applications, drones can be classified as civil, environmental, or military. Drones have a wide range of civil applications, including search and rescue operations for missing people, aerial photography, construction, recreation, inspection of electric power lines, manufacturing, transportation, logistic deliveries, crowd monitoring, surveillance, mining, and archaeology. One important application of drones is the delivery of medical supplies and medications in emergency cases. Drones are also useful in environmental sectors such as wildlife protection, crop monitoring, pollution control, mountain inspection, and land and water surveillance [1,2]. Drones are also used in scientific investigations, such as oceanic and cyclone monitoring in areas that are unreachable to humans. Drones were first used for military activities such as intelligence gathering, spying, military surveillance, and object tracking, but they have since also been used for civilian and environmental purposes. In the military sector, drones are applied in war zones, to combat aircraft, spying, border surveillance, attack and missile launching, and other use cases. There are numerous drone applications with diverse needs and goals, making it difficult to categorize aerial networks into specific application domains. Further detailed discussions on practical applications and case studies of drones can be found in [3,4,5,6,7,8,9,10,11]. Moreover, numerous Fifth Generation (5G)-related applications are emerging with the development of the new cellular technology, as indicated by 3GPP [12,13,14].

Drones have been recently included as User Equipment (UE) in the cellular architecture. The control link contains two major components: a point-to-point connection between the drone and the person maneuvering it, and a link that establishes a cellular network connection between the drone terminal and the Ground Control Station (GCS). Drones can also serve as ABSs in the sky to serve UE at specific locations. When drones are used as ABSs, they can support the connectivity of genuine terrestrial wireless networks such as broadband and cellular networks. The advantage of using drones as ABSs compared to conventional ground stations is their capability to alter their height, avoid obstacles, and improve the probability of creating Line-of-Sight (LoS) communication links for terrestrial users. Due to their unique properties such as flexibility, mobility, and adaptive altitude, Drone Base Stations (DBSs), can efficiently complement current cellular systems by providing supplementary capacity for hotspot locations. They can also offer network coverage in unreachable rural areas. Multiple linked drones can be used in certain situations where a single drone is incapable of delivering services provided by the drone network.

Another significant application of drones is their integration with the Internet of Things (IoT) [14,15,16,17,18,19]. IoT devices typically have low transmit power and may not be able to communicate over long ranges. Drones can also be used in surveillance scenarios, which is a key requirement for IoT. In cities or countries where towers and complete cellular infrastructure are expensive, drone deployment will become extremely beneficial since it eliminates the need for such costs. The conventional cellular architecture may be significantly altered to enable the application of drones in different service scenarios.

Various field tests have been conducted by several communication companies such as AT&T, China Mobile, Ericsson, ZTE, LG, Nokia, and Qualcomm [20,21,22,23,24]. Due to spectrum availability concerns, current investigations are underway using Wi-Fi, 802.15.4, and remote-control channels [10,25,26]. Other existing technologies have also been analyzed for wireless drone support such as 802.11, 802.15.4, Third Generation (3G)/Long-Term Evolution (LTE), and infrared. The authors in [27] examined the issue of drone interference in the context of adopting drone communications in the cellular infrastructure. Cell coverage and drone support have also been explored in the literature. However, extensive studies are still required.

Despite the potential prospects of drones, a range of practical challenges must be overcome to effectively apply them in each networking application. For instance, when using drone BSs, the most critical aspects to consider are performance characterization, drone implementation in optimal Three-Dimensional (3D) environments, wireless and computational resource management, flight time, trajectory optimization, and network planning. Handling channel modeling, low-latency control, 3D localization, and interference management are also key challenges in the connected drone concept. Among these challenges, efficient mobility (handover) management is a significant factor that must be addressed for drone BSs and drone UE scenarios [28]. To ensure smooth and reliable connection services while users are mobile, a secure connection must be established in addition to an efficient handover process.

Handover technology is the method of maintaining a continuous connection when a user moves from one cell to another without disrupting service. Serving signal level reduction, load balancing, and high error rates are among the factors that lead to the formation of handover actions. When one or more of the aforementioned factors reach an undesirable level, the connection must transfer to a suitable alternative cell for more reliable, stable, and seamless service. Although this process regularly occurs, it creates many challenges when the UE is a drone.

Several challenges must be overcome to manage handovers in mobile networks. System complexity increases with drone implementation due to their unique features. The drone’s flight may be controlled via LoS paths, even though the interference scale is greater than that in conventional terrestrial networks. Compared to the ground UE, the drone UE has a lower coverage probability since its antenna is tilted downward and the drone’s interference is overpowered by LoS [29,30]. Due to the higher speed of drones compared to that of the ground UE, the handover rate is comparatively higher. Since drones are supported by the sidelobe of the terrestrial antenna, many handovers will probably occur [31]. Consequently, the Quality of Service (QoS) will noticeably degrade [32].

Handover of drones must be professionally and expertly managed in terms of the techniques used to address handover challenges compared to current handover management in terrestrial UEs. Techniques and algorithms employed in terrestrial UEs may not be suitable for drone network applications due to their distinctive features. The key objective for using such methods is to deliver high-quality service and reliable communication while maintaining seamless handover between drones. Solutions have been investigated in several related works, but many challenges still remain. The provided algorithms are for both scenarios: drones acting as BSs and drones serving as UEs. The former scenario is under examination using the previously suggested algorithms. Drone BSs are assessed in two separate movement scenarios: drone BSs travelling in random directions at the same constant speed and drone BSs moving at various constant speeds.

In future mobile networks, node movement prediction is a key recommended technique for enhancing drone network service. Many contemporary methods are based on distance measurements and projections. Machine learning-assisted studies have been developed to support drone networks in acquiring certain patterns. This will enhance the performance of handover management, such as in [32,33].

This survey paper contributes to the target of providing a comprehensive and deep-focus review of handover management for connected drones in future mobile networks. The work provides a more focused review on drone networks, mobility management for drones, and related works in the literature. To illustrate how conventional technology functions in cellular-connected drones, the LTE system, common mobile ad hoc networks (MANETs), vehicular ad hoc networks (VANETs), and IEEE 802.11 standards are employed. Since drone networks are susceptible to frequent handovers, a variety of different handover strategies are extensively addressed to further proceed in making drones a viable alternative to ground BSs or UEs. These strategies address underlying issues without jeopardizing the performance of communication systems. Due to their mobility and flexibility, drones are preferred in a wide variety of applications. This review addresses methods for developing such applications while maintaining seamless handover management. The main contributions of this article are as follows:Providing a brief introduction to drone networks and connectivity requirements for drones and, more specifically, handover management in drone networks.Highlighting and discussing the main challenges facing the implementation of connected drones. The main focus is on the handover challenges that influence the mobility of connected drones in mobile networks, including the discussion of 6G and beyond in further detail.Summarizing and discussing the previous conducted research that has mostly focused on mobility management for connected drone networks, including performance, network operation, and connectivity issues.Discussing the key significant future research directions for connected drones. This includes mobility management, energy efficiency, machine learning, deep learning, IoT, MANETs applications, VANETs applications, new cellular technologies, security, and Mobile Edge Computing (MEC) with drones.

The analysis and discussion of this survey study indicates that, by implementing intelligent handover schemes that are based on machine learning, deep learning, and automatic robust processes, the handover problems and related issues may be reduced significantly as compared to traditional techniques.

The rest of this paper is organized as follows. Section 2 presents a brief description of the drone network. Section 3 discussing the connectivity requirements for drones. Section 4 reviews the handover management for efficient drone networks. Section 5 discusses the current handover challenges that must be addressed. Section 6 examines previous research regarding drone networks, mostly related to mobility management, performance, network operations, and connectivity issues. Section 7 highlights the most significant future directions. Section 8 presents the conclusion of this work. Table 1 lists and describes the abbreviations used in the text.

## 2. Drone Networks

Drones connected to mobile networks play a key role in enabling a wide range of services throughout various fields. The necessity for steady communication links while moving is a major challenge that must be thoroughly investigated. Therefore, this section provides a brief background on drone networks connected to mobile networks.

### 2.1. Drones Applications in Mobile Networks

Drone usage has dramatically risen in recent years due to their contribution to a wide range of solutions in a variety of fields, as illustrated in Figure 1. Drones have unique characteristics such as high mobility in 3D space, autonomous operation, and flexible distribution. This makes them attractive solutions for numerous applications including civilian, general safety, Industrial IoT (IIoT) platforms, protection, defense sections, cyber-physical systems, and atmospheric and ecological observation [10,34,35]. Drones can be extremely useful in a variety of civic and business applications when combined with 6G communication service, as illustrated later.

Drones can effectively provide wireless communication services by acting as BSs or as UEs in the sky. Drones can establish multiple connections, such as, Drone-to-Drone Networks, Drone-to-Ground mobile serving networks, Drone-to-Ground mobile users, and Drone-to-Satellite Networks. Ref. [36] examined drones serving as BSs in HetNets. The study focused on a solution for improving the wireless coverage of terrestrial UEs. This enables drones to be used in areas where communication is unavailable. Moreover, they can assist networks by acting as data relays between UEs and BSs. Drones are flying platforms that support adaptive height. Most emerging applications demand safe and reliable wireless communication systems with highly low-latency and well-organized information exchanges with the BS [5]. In current applications, drones are often equipped with specialized communication equipment or sensors to provide various services such as low-altitude surveillance, logistical applications, post-disaster rescue, and communication support.

Two primary work directions are thoroughly examined: the integration of drones in an appropriate cellular network scheme for smooth service, and universal connectivity for specific use cases. As seen in Figure 2, this integration allows drones to serve in three broad directions. The systemic and technical issues that arise as a result of this integration must also be analyzed.

### 2.2. Drones’ Connectivity

Ensuring smooth, reliable, and continuous connectivity for drones is one of the major challenges that faces the implementation of drones over mobile wireless networks. To provide reliable connectivity of connected drones, numerous solutions have been proposed for connectivity and networks’ management. For example, the authors in [37,38] proposed a machine learning method and massive Multiple-Input Multiple-Output (MIMO) design to enhance connectivity and security of connected drones, respectively. Ericsson has also introduced a control scheme, as demonstrated in Figure 3. The system manages drone flight administration while simultaneously coordinating with a manned aircraft management scheme. Additional information such as weather forecasts is fed to the drone’s control system.

A drone identification mechanism is implemented to identify, track, command, and control drones and drone fleet operators. Authentication and authorization processes are crucial for providing a secure communication. The system enables vehicle-to-vehicle (V2V) communication while avoiding collision. Wireless connection is necessary to establish the V2V communication and simultaneously provide network management. Connectivity can be accomplished over licensed or unlicensed spectrums. The former can be established via satellite communication or by utilizing ground cellular networks, which is generally more preferable.

The authors in [39] introduced a massive MIMO based on conjugate beamforming to provide more reliable connectivity to cellular networks. In [40], a system was designed with directional antennas for the drone BS to lessen the aerial UE’s LoS path. However, the works of [39,40] only assumed static hovering drones without considering the integration of cooperative communication, even though it leads to reduced levels of inter-cell interference. The framework presented in [29] utilized the Coordinated Multi-Point (CoMP) of maximum ratio transmission to enhance the Signal-to-Noise Ratio (SNR), thereby improving the cellular connectivity of aerial UEs. The network consists of several separate clusters in which BSs collectively provide services to one of the aerial UEs by utilizing CoMP communication. Two different situations are present: hovering and mobile drones.

### 2.3. Drones in 4G/5G Networks

Current Fourth Generation (4G) and 5G New Radio (NR) technologies used in autonomous vehicles for Vehicle-to-Everything (V2X) communication may be suitable for drones’ communication. The 5G NR can connect autonomous vehicles and infrastructures via side links [41], enabling Non-Line-of-Sight (NLoS) visibility and predictability for further traffic control and autonomous driving improvements. Since wireless networks are designed especially for ground mobile users, the usage of 4G and 5G NR may provide network connections such as UAV-to-UAV (U2U) (Drone-to-Drone) and UAV-to-Infrastructure (U2I); however, these do not guarantee full network coverage. Drones can also be used as BSs to provide 4G and 5G services in remote environments with limited coverage caused by natural disasters [42]. Existing 4G and 5G NR terrestrial networks are fixed at a specific location and can support ground users or vehicles traveling along predetermined routes. Fourth and Fifth Generation-NR systems can be utilized to provide communication for ultra-low-altitude drone networks using U2U and U2I modes. However, they may have coverage and other mobility issues, as discussed in the challenges section.

### 2.4. Drones in 6G Networks

Drones can be extremely useful in variety of civic and business applications when combined with 6G communication service that allow for smart automation and the integration of Artificial Intelligence (AI), paving the way for new services such as ultra-smart cities and Internet of Everything (IoE), Extended Reality (XR) (including Augmented Reality (AR), Virtual Reality (VR), and Mixed Reality (MR)), autonomous connectivity (such as autonomous vehicles), Wireless Brain Computer Interaction (WBCI), and AI-based services [43]. Sixth Generation is expected to offer 100 times the wireless connectivity and multiple times the performance of 5G. The most significant innovations that will drive 6G are satellite connectivity, drones, connected intelligence with machine learning, the terahertz (THz) band, Optical Wireless Communication (OWC), wireless power transfer, and 3D networking [44,45].

For air communications, 6G can overcome a number of limitations associated with previous generations of wireless communications [46,47,48,49,50,51,52,53]. Sixth Generation communications integrates a non-terrestrial network with 3D connectivity and ubiquitous AI-based services in 3D space, making it suitable for air communications. Sixth Generation technologies will provide seamless connectivity, high-accuracy positioning, ultra-high bandwidth, and real-time remotely controlled features in high-density aerial vehicle scenarios. Although drones may be affected by the utilization of terahertz (THz) bands due to high path loss and small coverage, the integration with satellite networks may solve the issue.

### 2.5. IoT-Equipped Drone Networks

The IoT network is the system of connecting everything around us to the Internet. The focus has recently shifted to drones. IoT supports numerous applications that drones can provide in addition to additional services. However, hindrances are present in turning drones into “flying IoT”, such as the large amounts of data needed for some applications and the communication mode selection in both LoS and NLoS. A trade-off exists between cost and efficiency since wireless communication has bounded accessibility while satellite communication, by comparison, is more expensive.

The Internet of Drone Things (IDT) is an emerging technological innovation with the potential to revolutionize AI computing and big data analytics. Ref. [54] presented a novel approach to detecting contagious disease pandemics based on AI-enabled IDT infrastructure and blockchain technology. The proposed system captures real-time geo-located data from various sensors on each drone and creates a unified IDT database for combined computational processing, storage, and retrieval. Algorithms are then employed for the identification of disease outbreak hotspots based on multi-source surveillance data collected from drones by combining visual meaningful images extracted from video streams with deep learning approaches. The combination of AI and Blockchain provides an efficient way to obtain authentic, reliable, and secure information about the contagion situation. As the IoT evolves, there is a growing need for new approaches to addressing IoT security, privacy, and scalability challenges. The authors in [55] introduced a federated learning-based Blockchain-embedded data accumulation scheme for remote areas where IoT devices encounter network supply shortages and potential cyberattacks. The proposed model consisted of a two-authentication process that validates requests first with a cuckoo filter, then with a timestamp nonce. Hampel filters and loss checks are used to ensure secure accumulation. Finally, model training is carried out in a suitable environment, and the results validate the possibility of the introduced model.

## 3. Connectivity Requirements for Drones

Drone implementation confronts a number of challenges; one of the most significant is the connectivity. Drones’ connectivity is more complicated than that of terrestrial UEs because of their characteristics. Drones, for example, have a higher mobility than UEs, resulting in a huge variance in the Reference Signal Received Power (RSRP). Connected drones may continue switching the connection link from one cell to another. As a result, connection between drones and the serving network may be quickly lost. To address this issue, several studies have been conducted throughout the literature, in which several key requirements for drones’ communication have been discussed. Thus, in this paper, the key requirements for drones’ communication, and its capabilities and 6G expectations, are discussed in the following.

### 3.1. High-Accuracy Positioning and Seamless Connectivity

Drones flying in multiple levels of airspace require precise localization and seamless connectivity, which are both required for network planning and implementation. A secure connection and extensive network coverage ensure seamless connectivity while the drones are flying autonomously. Covering a wide range of altitudes and maintaining reliable communication is a significant challenge for 4G/5G cellular networks. Sixth Generation integrates radar technology for high-accuracy localization and positioning. The development of dynamic maps and 3D positioning in the sky using a variety of high-tech sensors allows for high-accuracy positioning of drones. Multi-Level Networks (3D) composed of ultra-dense heterogeneous networks in 6G can boost the number of connected drones in high-density ecosystems by roughly 10^7^ devices/km^2^, which is 10 times higher than the connection density in 5G. A standardized, high-quality, and reliable cellular connection with extensive 6G coverage provides robust, cost-effective, and seamless connectivity beyond visible LoS. The high-capacity backhaul connectivity provided by the high-speed OWC system allows for the transmission of massive amount of drone traffic data.

### 3.2. Remote and Real-Time Control (RRC)

RRC depends on real-time flight progress reports from drones, including geo-coordinates and devices status. RRC enables a remote controller to release real-time command and control instructions. To allow remote control and tracking of the drones, specific data rates and latency criteria must be satisfied. With 6G, several drones can operate autonomously (i.e., autonomously in beyond-visible LoS). Sixth Generation connections integrated with satellites can provide communication over unlimited distances and provide near-instant control with a latency of less than 1 millisecond. If drones have 6G connectivity, they can be controlled from anywhere in the world using the Drones Traffic Management (UTM) system.

### 3.3. Multimedia Transmission

Some UAV-based systems handover data to ground stations, such as live multimedia/video streaming or data analysis, in order to save time. Advanced multimedia services such as truly immersive XR, 3D holograms, and 360-degree ultra-high image/video quality shoots (4K and 8K videos) must be eventually realized in the future. Furthermore, XR experiences such as AR, VR, and MR services necessitate higher data rates at higher Gbps levels. The 6G network can meet a high-bandwidth data connection requirement in the UTM. A sufficient bandwidth must be ensured for the improved data transmission capabilities that come with 6G technology, so that the drones do not constantly drop the connection and can transmit high-quality live videos. Sixth Generation is expected to deliver a data rate of up to 10 Gbps to support multimedia transmission [56].

### 3.4. Identification and Control of Aircraft

Due to the high volume of drones, the use of automatic dependent surveillance broadcast (ADS-B) for recognizing commercial aircraft may overload its frequencies in the future. As a result, a new identification technique is required. The remote identification data can be used in conjunction with 6G, and act as license plates in the same way as license plates in vehicles. Radio waves are used to transmit the remote identification. Aircraft registration, identification, tracking, and regulation all necessitate reliable cellular network connectivity. Drones’ traffic conditions can be detected and measured by actively monitoring drone locations and route details, and early recognition of geo-fencing and potential attacks can be identified accordingly. The UTM ecosystem provides Low Altitude Authorization And Notification Capability (LAANC) for drones, allowing drone operators to access controlled airspace near airports via real-time verification of airspace authorization below authorized altitudes and management of dynamic geofencing [57].

## 4. Handover Management for Drone Networks

Drones will serve various environments and be a significant part of future mobile networks. However, handover management will be a critical matter that must be addressed in future networks. Accordingly, this section highlights handover management in drone networks.

### 4.1. Handover in Drone Networks

The handover performance is a common assessment in cellular networks since it is a good indicator for demonstrating the efficient mobility techniques. Handover, or handoff, is a key technique in mobile networks that allows a UE to switch its connection across BSs while on the move. Handover with drone networks has become a more significant matter because the connected drones move in the sky faster with different characterizations. Depending on the functionality of drones within the network, one or several drones may be needed to provide network access services to specific terrestrial users. Drones may also serve as UEs and receive service from ground BSs or from satellite networks. Since a drone’s operation is restricted by its power, coverage, mobility characterizations, and serving network traffic, handover will be increasingly required. The handover (handoff) process is crucial for the continuation of a connection, imposing only a short delay [58]. Furthermore, the drone network remains highly dynamic since mobile aerial vehicles and the radio environment are different compared to ground users due to several factors, such as high altitude [26,59]. The traditional handover control systems in MANETs and VANETs must be altered to be suitable for drone networks. In MANETs, the commonly utilized handover techniques lead to constantly separating or merging network nodes [60]. Several architectures for drone traffic control systems have been proposed. For instance, NASA and the Federal Aviation Administration (FAA) proposed the UTM scheme [61]. The European Union is also developing U-space, which contains a set of guidelines and services [62].

### 4.2. Handover Decision Algorithms

A variety of handover decision-making algorithms are used in cellular networks, such as RSRP, Received Signal Strength Indicator (RSSI) of the Serving Base Station (S-BS), the Signal-to-Interference-Plus-Noise Ratio (SINR), mobile movement speed, distance between the UE and BS, limited capacity of BSs, weight functions, cost functions, fuzzy logic control, and machine with deep learning technology. The same handover decision algorithms can be used with drones, but the performance will differ due to the different characterization of drones [63,64,65,66,67,68,69,70]. Moreover, the requirements of 6G technology will be ultra-high compared to those of the previous mobile systems. This also creates the need for more robust, efficient, dynamic, and smart handover decision algorithms for drones’ networks. Several studies have been conducted in the literature that deal with this matter.

For example, the authors in [67] created a method for establishing drone connectivity with IoT. The model architecture consists of two main nodes: the sensor node and the data processing node. Two different modes of communication are utilized: Wi-Fi and satellite communication. The handoff was performed based on several parameters: network accessibility, RSSI, QoS, cost of data transmission, and network performance. If one of the previous criteria indicated that the Wi-Fi interface is not the optimal choice, vertical handover is performed to switch to the satellite communication mode. If neither of the interfaces correctly operate, buffering is then performed to avoid packet loss until one of the interfaces becomes available.

The authors in [23] investigated a method that analyzes the impact of heterogeneous movement Device-to-Device (D2D) drone-supported Mission-Critical Machine-Type Communication (mcMTC) in 5G. Due to the rapid increase in the use of IoT systems, mcMTC’s role has become extremely significant. Therefore, fulfilling these extensive requirements is necessary. The paper examined the influence of various movement patterns on heterogeneous users. The study verified that, as long as alternative connectivity options are in use, availability will increase. The WINTERsim simulator was applied for the evaluation.

The impact of a heterogeneous device’s movement is based on the multi-connectivity options, which introduce three measured cases: vehicular connection, manufacturing automation, and city communications. The UEs included in the multi-connectivity system can utilize D2D, cellular, and drone-supported connections. Ref. [68] proved that low and limited mobility of the device has no effect on the connection availability and reliability. Since the packet sizes are diverse, the use of D2D-assisted communications and drones greatly enhances reliability and data rates. In contrast, performance degradation was detected for cases where movement was high.

### 4.3. Handover Types

Handover in cellular networks can be classified into different types, based on technique, network type, network management, operating frequency, and scenario. ***For example,*** handover can be classified into two main handover technique types: hard and soft handover techniques. The hard handover requires the UE to terminate the connection from the serving BS before it switches to the target BS. The soft handover imposes a more gradual connection termination, simultaneously maintaining a connection with two or more BSs for a short period of time [69]. The drones’ network can apply two different handover techniques depending on the mobile communication technology.

Handover also can be classified into different types based on the technology of the serving and target networks. The two main types are horizontal handover and vertical handover. In the horizontal handover, the access points use the same technology and the network interface remains unchanged. In vertical handover, the access technologies are different from each other, and multiple network interfaces are employed. For instance, the user switches from the terrestrial cellular network to satellite technology, as illustrated in Figure 4.

Furthermore, handover in cellular networks can be classified into three methods depending on the network management system: (i) Network-Controlled Handoff (NCHO), (ii) Mobile-Assisted Handoff (MAHO), or (iii) Mobile-Controlled Handoff (MCHO) [70,71]. The handover control system is extensively described in [72]. For example, if the recipient signal is the mechanism triggering parameter, two handover scenarios will occur: absolute or relative. The former occurs when the serving BS signal strength becomes lower than a pre-defined threshold value, whereas the latter occurs when the serving RSRP is lower than that of the target BS. The relative handover technique may cause handover to occur earlier than needed yet provides higher quality. Absolute handover, however, causes what is referred to as the “ping-pong” effect. This phenomenon occurs from frequent variations in the RSRP value, prompting frequent handovers. These various handover types can also be applied with drones’ networks.

### 4.4. Handover Procedure in 5G

The handover procedure is a significant process that consists of different steps, algorithms, and techniques to enable UEs to switch connections from one cell to another. The procedural steps differ from one technology to another. The same procedure used for the terrestrial UE can work with drones; however, it does not guarantee efficient handover performance since the characterization of drones is different. This subsection provides a brief description of the handover procedure for one handover system scenario that may occur, as illustrated in Figure 5 (as an example).

The 5G handover process is closely similar to LTE-Advanced system with some further enhancements. The Access and Mobility Management Function (AMF) conducts the responsibility of the Mobility Management Entity (MME) [73]. The User Plane Function (UPF) is the same as the Serving Gateway (SGW). The handover procedures are listed as follows:The UE periodically sends the measurement report to the S-BS.The S-BS configures the measurement procedure of the UE.Based on the measurement report, the S-BS makes the switch decision, and the handover request is then received by the Target Base Station (T-BS).The T-BS replies with an acknowledgment to the S-BS based on its resources.The handover is initiated, and the T-BS supplies the UE with the necessary information, connecting it to the target cell.The UE receives uplink allocation and timing info sent from the T-BS.The T-BS updates the AMF for UE cell alteration, the UPF is updated by the AMF for the UE, the path of the UE is updated by the UPF, then the AMF notifies the T-BS for path update.The S-BS is updated by the T-BS for the completion of the handover.

Another way of categorizing handover is based on whether the UE controls or assists in the process. A handover in which both the network and the UE are involved is known as a hybrid handover. These categories have been investigated for mobile Internet Protocol (IP) networks and VANETs, but only a few studies are currently available for drone networks.

## 5. Handover Challenges in Drone Networks

Drones connected to cellular networks will be a vital infrastructure that offers a wide range of services in various environments. The necessity for stable communication during their movement is a major challenge that must be emphasized. Several challenges arise with the implementation of connected drones due to tier connectivity and movement characterizations. Handover issues would result in high handover rates [74,75,76], which would lead to a large ping-pong effect [77], or a high rate of Radio Link Failures (RLFs) [78] or Handover Failures (HOFs) [30,79,80,81,82,83,84,85]. RLFs and HOFs are both significant key performance indicators in mobile networks. Both may increase due to the high speed of mobile users, the suboptimal settings of handover control parameters, inefficient handover decisions, and other related factors.

### 5.1. Drones’ Connectivity

Ensuring smooth, reliable, and continuous connectivity for drones is one of the major challenges faced in the implementation of drones over mobile wireless networks. Drones flying in multiple levels of airspace require seamless connectivity, which is required for network planning and implementation. An extensive network coverage ensures seamless connectivity while the drones are flying autonomously. However, covering a wide range of altitudes and maintaining reliable communication is a significant challenge for 4G/5G cellular networks. This is due to different factors, such as the fast movements of drones, the different trajectories, high levels of interference due to the LoS connections, and movement in 3D. Moreover, drones move in the sky faster than UEs, resulting in a large variance in the RSRP. This will lead the connected drones to continue switching the connection link from one cell to another much more than in the terrestrial UEs. Moreover, this large variance in the RSRP may lead quickly to connection loss between drones and the serving network. Moreover, the fast growth in the use of drones will require high-capacity backhaul connectivity to ensure their reliable and smooth connectivity. Furthermore, the Multi-Level Networks (3D) composed of ultra-dense heterogeneous networks in 6G can boost the number of connected drones in high-density ecosystems by roughly 10 times compared to the connection density in 5G. This also negatively impacts the drone’s connectivity. Furthermore, aircraft registration, identification, tracking, regulation, and control all necessitate reliable cellular network connectivity. Drone traffic conditions can be detected and measured by actively monitoring drones’ locations and route details, and early recognition of geofencing and potential attacks can be identified accordingly. In addition, the massive growth in the use of drones, IoT applications, U2U, V2V, V2X, M2M, D2D, AR, and all the other connected devices will negatively impact connectivity. Thus, the future mobile networks will provide high-quality and reliable cellular connection with extensive coverage, and provide robust, cost-effective, and seamless connectivity beyond visible LoS.

### 5.2. Drones Challenges with 4G and 5G Networks

The 4G and 5G-NR systems can be utilized to provide communication for ultra-low-altitude drone networks using U2U and U2I modes. However, they may have coverage issues, whereas drones travel in 3D and at much higher altitudes, i.e., from more than 150 m to 2 km, further overcoming mobility challenges. Drones, by comparison, are able to move randomly and discontinuously in any 3D direction in space at very high speed. Although 5G can handle the 2D mode, it may have obstruction issues that make the 3D mode difficult to handle. Due to the use of directional antennas in the BS, 5G has limited connectivity and necessitates frequent handovers for high-mobility drones. To cover high-density drones in the sky, additional antennas must be installed throughout the BS, which may be costly. Fifth Generation connectivity is incapable of handling dynamic handover management or providing seamless connectivity with path planning in a high-mobility scenario, such as cellular V2X, both of which are essential for autonomous drones flying in the airspace [86]. In high-density and city air mobility scenarios, latency, collision detection, and navigation are crucial, necessitating energy-aware deployment, ultra-High Speed with Low-Latency Communications (uHSLLC), and effective channel models for drones’ communication. Considering the opportunities of developing technologies and services for the next decade, there is a significant need to move beyond 2D infrastructure coverage to fully 3D native services.

### 5.3. Interference Probability

At ground level, handover is generally performed on the distance basis in some cases, indicating that terrestrial UEs receive service from the closest BS. Drones have fewer barriers that may block the signal due to their elevation as compared to ground UEs. Thus, the number of LoS links is greater than that available to terrestrial UEs, as illustrated in Figure 6. The drones may have direct links to non-serving BSs, raising the interference probability. This will create interference issues in the drone’s network. This may also lead to the increasing probability of handover execution, especially if the handover decision algorithm is taken based on the RSRP or SINR level.

### 5.4. Sidelobes

The main lobe may not serve drones properly since the antennas are not omnidirectionally vertical, as illustrated in Figure 7. Although research efforts have been made, the issue remains unresolved. It has been shown in [87] that the effect of sidelobes is mitigated at high altitudes when drones are exposed to free-space propagation conditions. Sidelobes are part of the far field pattern of a directional antenna, transmitting undesirable radiation in directions other than the main one. Since sidelobes have a lower field intensity than the main lobe, terrestrial users connect to the main lobe by tilting their antennas downwards. Drones may be prone to unwanted sidelobes since they fly at high altitudes. The sky may not be fully covered by the sidelobes of BSs, resulting in no sky coverage and subsequent link failure. This also contributes to further increasing the handover probability.

### 5.5. Security and Privacy

Drones’ networks are also at risk to various types of the privacy and security threats. Therefore, it is essential to protect upright sky networks from any related privacy and security threats. Various survey and research studies have been conducted and provided a vital basis for considering the drone threats that need to be addressed [88,89,90,91]. However, no study has provided an optimal solution for the existing issues. More studies related to the privacy, security, security level, privacy threats, secured architecture, types of attacks, and more efficient attack mitigation techniques still need to be conducted.

### 5.6. High Mobility Speeds

Controlling the movement of drones while in flight is one of the most challenging aspects of drone operation. Drone movement in the atmosphere is extremely complicated and difficult to control. The high mobility and arbitrary acceleration of a drone, for instance, causes sudden and instantaneous variations in the obtained signal frequency. The rapid variation in the received signal also increases the probability of handover and other mobility-related issues. Therefore, current handover techniques may not be sufficient for drones.

### 5.7. Handover Self-Optimization Functions

Another issue result from the Handover Self-Optimization Functions that deal with handover control parameters, such as Mobility Robustness Self-Optimization and Load Balancing Self-Optimization functions. To date, several handover control techniques have been developed in the literature to optimize handover control parameters; however, existing techniques may not be able to work efficiently with drones. The existing techniques were designed to serve terrestrial users, which have very different mobility, operations, and traffic features. The key distinctions between drones flying in the sky and terrestrial users produce significant challenges. Moreover, most of the literature has been developed with the previous mobile networks, whereas the work on 6G networks is still in its infancy. Existing handover Self-Optimization functions still pose issues that require further enhancements.

### 5.8. Handover Decision Algorithm

The handover decision algorithm is another key challenge in drone networks. Numerous types of handover solutions exist for managing handover decisions, such as distance, RSRP plus distance, RSRP, route information, SINR, loads, mobility speed, cost functions, and machine learning. The algorithms based on RSRP are generally less complex but are also less precise. A vital feature of algorithms is that various standards can be applied for the handover decision-making procedure. This further increase computational complexity but enhance efficiency and accuracy. Most existing algorithms are for previous generations, which are completely different in terms of specifications compared to the current generation. Drones have distinct characterizations, making existing algorithms inefficient. Further analysis and enhancements are still required.

### 5.9. Handover Failures

The HOF may occur due to the fast movements of drones and the delay in the handover initiation or procedure with connected drones. Several works have been conducted to address this issue in mobile networks [74,75]. In [74], the authors proposed a distance-based handover algorithm for femto and macro cells to mitigate the HOFs and unwanted handover. In [75], the reactive handover method was used to delay the handover process until the UE loses connection from the previous BS or reaches the most predictable position. This technique minimized the overall handover performance. However, in the case of drones, further investigations that include various mobility speeds and system settings scenarios are still needed.

### 5.10. Handover Ping-Pong Effect

In connected drones, the handover “ping-pong” effect is more likely to occur than in the terrestrial case. This is due to various factors such as the fast mobile speeds of drones, suboptimal use of handover control parameters, the use of an inefficient handover decision algorithm, and the increase in LoS connection probabilities. The false activation of the handover process may further be a contributing factor. The combined factors cause repeated handovers. Moreover, if drones act as UEs, determining the relation between the drone and BSs will be another issue.

### 5.11. Radio Link Failures

The RLF is an alternative challenge that may increase with the implementation of drones as compared to terrestrial UEs. Both drone characterizations and inefficient handover techniques are factors that contribute to the increase in RLF. Handover optimization algorithms and handover decision techniques play a key role in controlling the occurrence of RLFs in mobile networks. These factors must be collectively considered. Several optimization algorithms have been proposed for robust distributed movement to mitigate the RLFs and HOFs by altering the offset parameters, such as Handover Margin (HOM) and Time-To-Trigger (TTT). However, the issue remains a challenge, especially with 6G mobile networks that will be characterized with high requirements and specifications.

### 5.12. Other Mobility Issues

Additional crucial parameters that determine network performance are QoS, bandwidth, power levels, coverage, use of high frequency bands, and latency [67]. Due to the high mobility of drones, drone networks are more vulnerable to frequent handovers. Conventional handover mechanisms will be ineffective. New techniques must consider the potential challenges that drone networks will face.

## 6. Related Works

Several research works were conducted on handover/mobility management in mobile networks [92,93,94,95,96,97,98,99,100,101]. Different classifications for these issues were reported and discussed in the literature; however, the focus on drones is still limited. Several studies concentrating on drone networks were undertaken to address various issues such as mobility management connectivity, IoT applications, and multi-user access control. This section divides the related works into subsections. The first subsection discusses the different solutions for handover/mobility management based on various concepts and techniques. The second subsection presents the proposed solutions for handover/mobility management based on machine learning technology. The third subsection presents the proposed solutions for handover/mobility management based on deep learning technology. Then, the fourth subsection presents different proposed solutions for various communication issues based on different proposed solutions.

### 6.1. Classical Based Techniques for Handover Management

In 2006 [102], the authors demonstrated that PMIPv6 enables two 802.11p systems to conduct reliable communication between each other. Since nodes require extra time to be ready to re-receive or re-transmit packets after handoff is performed, MIPv6 is generally not preferred. In drone networks, the fundamental concept is that the handoff procedure will include three nodes: the ground mobile UE, the drone that will go out of service, and the drone to which the UE will be connected to next.

In 2009 [103], Media Independent Handover (MIH) with the IEEE 802.21 standard was examined to address communication between layer 2 wireless systems and layer 3 IP networks. It provides handover support between IEEE-802 and non-IEEE-802 systems, such as mobile communication networks. It can acquire information from lower layers by utilizing media independent interfaces. This technology enables drones to communicate in an integrated fashion [104]. Software-Defined Network (SDN) methods that employ OpenFlow standards were implemented for nodes configuration. MIH mechanisms were also applied for handoff optimization in non-homogeneous wireless networks.

In 2012 [105], the authors employed the VIP-WAVE (vehicular IP) technology together with proxy mobile IPv6 (PMIP) to provide more efficient handover support than conventional WAVE systems. Fast mobile IPv6 (FMIPv6) and hierarchical mobile IPv6 (HMIPv6) were modified to present more advanced MIPv6 schemes that demonstrate better handoff performance. Mobile IP version 4 (MIPv4) and Mobile IP version 6 (MIPv6) are common protocols also employed in VANETs. Studies have compared PMIPv6 to MIPv6. The simulation results reveal that the former is more effective, especially when the 802.11p standard is employed.

In 2012 [106], the authors studied seamless horizontal and vertical mobility in the VANET environment. The handoff assessment for the cellular-connected drones’ network was analyzed, and the LTE-Advanced system was specifically implemented. As previously mentioned, both secure and reliable communication are crucial in the drone network. The IEEE 802.11 WLAN module was employed to provide the wireless link and exchange the control commands and sensor data. Since the drone models were implemented in 3D, IEEE 802.11 WLAN technology was incapable of supporting all network services while maintaining reliability and security during connections. Although preceding network architectures have significant characteristics that are mirrored in the drone network, since the underlying infrastructures were not designed for aerial vehicles, issues emerged such as high interference and insufficient coverage [107]. These challenges should be resolved so that existing cellular systems can be used [103]. Previous studies have indicated that the number of BSs available to drones increases with altitude; however, the handoff rate was not computed. The authors in the underlying paper discovered a relationship between the handover rate and altitude. The antenna was tilted downwards to provide service to terrestrial users. Not every antenna design is applicable in aerial vehicle networks. To demonstrate the hindrances that emerge with drone UEs, let us consider the following situation. Two BSs (BSA and BSB) are present. The drone at position P1 is served by BSB; however, it is near BSA. When it moves to position P2, it switches to BSA. It then switches back to BSB when moving to P3. Therefore, frequent handover risks will rise, regardless of the short distance between movements. This scenario is similar to that presented in Figure 7. The field measurements are crucial to provide further understanding of the LTE-connected drone network. An antenna tilt of 5–10 cm and a transmit power of 20 W was used. The drone maneuvers were either performed manually or autonomously using the Global Positioning System (GPS). The TCP protocol was also applied. The assessed parameters included the RSRP, the Physical Cell Identity (PCI), and the RSP quality of the radio and sensor data. To establish an efficient handover algorithm, one must consider the outcomes of this algorithm.

In 2015 [108], the authors conducted research on optimal coverage control for net-drone handover. The authors in this paper proposed a coverage decision algorithm, which aims to offer seamless handover and complete coverage for the connected drones’ network. The authors presented an algorithm based on the RSS. The algorithm adjusts the drone’s altitude and separation distance. The proposed coverage decision algorithm is evaluated in terms of success and false handover probabilities. The work was conducted by a simulation study in order to assess the performance of the proposed algorithm. The presented simulation results illustrated that the proposed algorithm is capable for drone networks. Regarding simulations, the algorithm performs admirably. However, a more accurate scenario must also be considered, considering the drone’s payload, BS radio range, etc. Furthermore, the coverage algorithm considers each drone’s RSS the same, which may not be readily applicable.

In 2015 [26], the authors highlighted the different technologies and protocols that can be applied in the drone network, and their corresponding performance. In VANETs, for instance, no field experimental works are available concerning the mobility issue by implementing the Wireless Access in Vehicular Environments (WAVE) system. WAVE communication includes both IEEE 1609.x and IEEE802.11p standards. Nonetheless, these technologies do not provide any solution to mobility challenges. VANET schemes are characterized with high dynamism due to the constantly changing system topology and mobile nodes. Since VANET is a subordinate group of MANET, the latter’s standards can also be applied to a VANET system [36]. In the RWP structure, the node’s motion is assumed to be random. Due to the constantly separating or merging topology elements, it is not possible to maintain existing paths or predetermine the best path. It is therefore necessary to reconfigure the system, which causes frequent handovers. The model’s performance and QoS deteriorate due to the delay and errors caused by the handoffs. Therefore, RWP is rarely used for mobile users or BS scenarios. The implemented network must eliminate the topographical limitations of nodes. Manhattan and Street Random Waypoint (STRAW) are two commonly employed systems. An additional challenge is that the density of VANET nodes must be equal to the number of nodes needed for the corresponding application. These obstacles hinder researchers from implementing a seamless handover scheme to provide reliable communication and enhance packet delay [109,110]. Unfortunately, there is a lack of research regarding handover schemes in VANETs and Wireless Mesh Networks (WMNs) that apply IEEE 802.11 standards [106].

In 2016 [111], the authors investigated a suitable orientation-based fast handover method to overcome the ping-pong effect for LTE-Advanced systems. They selected the T-BS based on the current load and Received Signal Strength (RSS). In [77], a Reduced Early Handover (REHO) technique was suggested to minimize both the ping-pongs and RLFs, achieving high energy efficiency while maintaining other performance parameters within appropriate limits. The finding in [55] also led to the outcomes of [56] where a fuzzy multiple criteria cell selection technique was used. This scheme considers the UE uplink conditions, resource block allocation, and selection criteria of the LTE’s conventional cell selection approach. This provides high reliability and minimizes both the HOF and ping-pong effect, thereby increasing throughput. The authors in [112] proposed a Handover Detection Self-Organizing Handover Parameter (HD-SOHP) scheme that relies on the reinforcement learning (RL) principle. This approach enhances the performance of UE mobility by maintaining low HOFs, ping-pong, and call drops. 

In 2016 [110], the authors presented an effective handover technique for drone network services based on 3D rather than 2D. This method was used to optimize network services by adjusting the drone’s height and distance. The optimum coverage decision procedure was assessed using seamless handover probability (*Ps*) and false handover initiation probability (*P_f_*). The altitude of each drone must be modified by considering physical restrictions to ensure the same coverage for each drone. A smooth handover can then be achieved. Various scenarios have been provided in terms of *Ps* and *P_f_* using simulations. The results indicate that the overlapped area’s vertical distance reduced when *P_f_* increases and *Ps* decreases. Frequent handovers are generally terminated, allowing the system to save the drone’s battery. This method guarantees a drone’s network optimization by determining the optimum overlapping area. Interference between drones can be reduced by setting the same RSRP for each drone. In contrast, several crucial issues are present in terms of optimum drone coverage. For instance, the minimum threshold level of a drone’s altitude must be changed if it is unable to fly at a low level. The RSRP level must also increase to guarantee a seamless handover when drones are affected by a change in climate such as rain and wind. Further considerations must also include the throughput rate and system reliability.

In 2017 [101], the authors extensively examined LTE-based drone network elements and their functionalities. The MME is crucial in managing handovers between the GCSs. Several MMEs are needed to control the handover process. The SGW and Packet Data Network Gateway (PGW) elements are used to manage the IP communication between drones and corresponding control stations. The Home Subscriber Server (HSS) or authentication center (AuC) has similar functionality as in the LTE cellular system. The control entity in the underlying drone system consists of four fundamental sectors: LTE sector, Drone-to-GCS link, GCS-LTE link, and Wi-Fi sector.

In 2017 [95], the frequent handovers between small-BSs and the load distribution were considered. Ref. [96] also presented another algorithm to overcome unnecessary handovers and signaling overheads for HetNets with massive small cell distribution. This was achieved by measuring the distance between small cells and the UE. The UE’s movement angle was considered to generate a short list of candidates that can be used to reduce signaling overheads and unwanted handovers.

In 2017 [92], the authors conducted research focused on handover management in software-defined ultra-dense 5G mobile networks. They proposed a Markov chain-based handover management strategy. This method selects and allocates the next best eNB to Open-Flow tables of the mobile node (before the real connection) while considering the available resource and transition probability approximation. Unlike the normal approach, this method reduces HOFs and delays to 21% and 52%, respectively.

In 2017 [113], the authors proposed an intelligent handoff algorithm for the drone network. The heart of the algorithm is a fuzzy inference mechanism whose functionality is based on the comparison of several input parameters. Initially, information collection is established in either the devices or BSs. The devices determine whether they prefer to remain connected to a specific BS or switch to a different one. The parameters that the handoff decision relies on are classified into two classes. The first class consists of network specifications such as the RSS, communication coverage, and radius. The second class consists of device specifications such as the altitude and UE speed. The reception signal and coverage have an inverse relationship since the coverage increases as the RSS decreases. Alternatively, the handover rate increases when the drone’s motion is rapid. An optimized algorithm was built to correlate handover with the drone speed. The reception signal decreases with the distance between the BS and drones. After information acquisition, the fuzzification process can then be established. This includes the normalization of input parameters and a linguistic variable (i.e., high, average, or low are appointed for the coverage, and high or low are selected for the speed). A pre-defined table consisting of fuzzy inference rules is the core of the handoff decision-making algorithm. A defuzzification mechanism for the parameters was then employed, such as Centroid of Area (CoA). The MATLAB fuzzy logic toolbox was used during simulations, in addition to a system consisting of three BSs and a single drone. For the drone’s motion, two different models were established: random and straight movements. A hundred simulations were conducted for several different drone paths and directions. Table 2 presents the outcomes of the algorithm. As demonstrated in the table, the number of handovers decreases in both scenarios (i.e., random and straight motions) when compared to conventional methods.

In 2018 [99,100], the authors addressed handover signaling minimization. A mobility management technique based on mobile user position tracking was introduced in [99] to obtain practical and smooth handovers. The UE transmits Sounding Reference Signals (SRSs) used for tracking its position by determining the arrived angle and LoS path. This method mitigates the handover signaling cost and achieves smooth mobility at the expense of increasing computational complexity. It was found that the anchor-based multi-connectivity technique supports a low handover rate and cost. With multi-connectivity, proper access points can be selected as handover anchors to allow the control plane, thus minimizing the handover rate. A mobility-aware user association mechanism for 5G (millimeter-wave (mm-wave)) networks has interesting features. It prevents frequent HOs between small-BSs and considers load distribution as proposed in [59].

In 2018 [114], the authors conducted study on mobility prediction in drone networks. They proposed a machine learning-based solution for classifying mobility based on predicted node locations in the near future. This system can be improved further because it can gain knowledge by itself. If this system is properly combined with routing protocols, it can assist in the prediction of future network topologies. However, the system ignores the fact that practical UAV networks have limited tracking resources and computational power.

In 2019 [60], the authors reported on and studied the cell selection and handover for drones connected to an LTE-A network based on real measurement data collected in a suburban environment. The presented experiments illustrate the impact of increasing flight altitude on the handover performance in terms of handover rate. The results showed that the flying drone at a typical height of 150 m is projected to switch the connection, i.e., execute handover, five times in each minute (as an average) as compared to only one handover process for the terrestrial users moving at a similar speed. From this study, it was concluded that more efficient handover techniques for connected drones are required in the planning and operation of future mobile networks.

In 2019 [39], the authors evaluated the handover performance in the 3D designed system. The simulation outcomes indicate that the altitude of aerial UEs, the distance between drone UEs, and the serving BSs all play a crucial role in the drone’s performance. Due to LoS interference and the sidelobes effect, the handover probability’s upper boundary is still comparable to that of terrestrial UEs. The CoMP transmission improves the performance of high-altitude drones. A 2D homogeneous PPP was used for locating BSs with the same transmitting power levels and height. Drones, both static and moving, positioned at higher altitudes than the BSs, have been considered. Separate clusters of BSs were created, each with a defined center distance. Two channel models were employed: high and low fading. The former considers both LoS and NLoS paths. A Nakagamim scheme was utilized for the low fading case. Down-tilted antennas and a rectangular layout for the antenna gains were considered so as to closely resemble real-life scenarios. The handover rate was assessed for both stationary and moving aerial UEs. Cauchy’s inequality and Gamma moment approximation were employed to compute the lower and upper limits of the handover probability for both cases. CoMP transmission proved to be quite useful since it is applicable in mobile drone-mounted BS scenarios.

In 2019 [59], the authors conducted experimental work for HO management investigation. This research focused on which parameters affect cell selection and HO management in UAV-UEs and how they affect them. More HOs will occur as altitude increases. When the altitude of the drone increases, the drone will also connect with more distant cells. Because of minor changes in RSRP values, cells change frequently. To fully integrate drones in 4G, 5G, and 6G networks, advanced solutions are required.

In 2019 [115,116], the authors conducted research on the fundamental analysis of drone cellular networks under the random waypoint mobility model. The work compared the performance of an ultra-dense millimeter-wave network architecture having the control and UE-plane with that of a previous architecture. The analytical framework was used to reduce the cost of handovers for a specific coverage area requirement. The handover costs and coverage area possibilities were found to be better than those of conventional schemes. The handover cost of the conventional scheme can further be decreased by adding more macro-BSs, which is preferable to increasing the number of small cells for the proposed scheme. The proposed scheme will be an advantage in the implementation of a 5G platform. However, further investigations with 6G mobile networks and drones are needed.

In 2019 [117], the authors studied mobility management for drone networks and they considered a stochastic geometry-based mobility model for drone cellular networks. Drone BSs were initially distributed according to the PPP and UEs movement based on a Random Waypoint (RWP) mobility model. The drone BS that serves a typical UE on the ground was chosen according to the nearby neighbor association rule. First, each drone flew around for a stable time interval, choosing a regularly random route, and then moving at a fixed speed for a constant distance. Again, the drone flew for a similar time interval in the new position until it traveled the same distance but in another random route.

In 2019 [115], a multi-tier 3D drone network was designed to compute the corresponding handover probability. The drone-mounted BSs provide services to terrestrial users. Both the horizontal position of drones and the terrestrial UEs’ locations are determined using the PPP models of different densities. The drone BSs have the same values of transmission power and path loss exponent. The channel power gain of the LoS path was also measured. The piers were positioned at equal horizontal distances from each other. In the traditional case, the handover between tiers is performed depending on the receiving power. The UE obtains services from the drone that offers the highest level of receiving power. The study’s model presented a different scenario: when all drone positions are of the same altitude, the horizontal distance of each is measured and the ground user is connected to the drone that provides the smallest horizontal distance. This represents the association criterion based on the probability that a terrestrial user is connected to a specific drone BS. The handover probability of a UE between multiple piers was also computed. Assuming that a user is connected to Drone-a of horizontal distance *d_A_* and another adjacent drone, Drone-b, has a horizontal distance *d_b_*, the handover process denoted by *HO_A_*_,*B*_ is initiated if d_B_ < d_A_. The handover probability can thus be measured based on the horizontal distance and on the direction of motion. The association process depends on the drone density. If the density is low, the association probabilities of tiers are almost the same. However, when the altitude of tiers is higher, the handover probability drops since the association process does not maintain equal value for each tier behavior at high altitudes. The handoff process therefore occurs more frequently. The authors provided significant insight concerning the handover probability in a drone network. An optimal density value was also presented, which is crucial to avoid frequent handovers.

In 2019 [118], the authors conducted experimental work for understanding the performance of UAV networks. This study conducted experimental works based on successful and failed HOs, the RLF number, and the rate of ping-pong HOs to analyze changes and challenges in the radio environment. The results of the experiments show that the HO rate increases with speed, as expected. Furthermore, because the HO procedure takes time, RLFs frequently occur when signal strength drops due to nulls between the antennas’ lobes.

In 2019 [119], the authors have conducted study on route-aware handover enhancement for drones in cellular networks. They presented a route-aware algorithm. This algorithm is based on path information, which is used to optimize the network using flight path data. The HOF can be reduced by 5 to 24 times for aerial UEs of varying speeds. The results can be improved further if the radio link quality is presented at a finer granularity and the estimation accuracy is improved.

In 2019 [120], the authors studied mobility-driven routing in autonomous drone logistics networks. They proposed novel DTN optimization of packet routing. This algorithm improves packet routing based on priority, time to live, and power consumption constraints. The packets are weighted according to their priority, time to live, and power consumption. If the packet’s time to live runs out, it will be dropped off. Furthermore, the algorithm’s output is a new path, which is executed if it is shorter than the maximum length that the drone can fly. When the opposite is true, the path is removed.

In 2019 [121], the authors studied the effects of mobility uncertainties on wireless communications between flying drones in the mm-wave/THz bands. They examined the use of mm-waves and THz band communications in drone networks. During the small-scale mobility, the performance decreased by about 2.5% after 5 s without beam alignment. In this case, a continuous disconnection is observed. In general, small and large moves degrade performance by up to 50%.

In 2019 [122], the authors studied the location module for a 5G base station to support mobility management of drones. They proposed a location module for tracking. This proposed location module can be integrated in Sensor Gateways and 5G BS to monitor UAVs and learn about their state while they are moving. Implementing advanced machine learning can enable additional services such as address discovery, navigation, and product delivery.

In 2020 [28], the authors proposed a method for estimating the handover probability for drone-mounted BSs. The handover probability of two different scenarios was assessed: drone BSs moving with the same constant speed along a straight line of constant height in random directions, and drone BSs moving in different constant speeds. In the first scenario, a similarity was present between the spatial distribution of the mobile drone BS and static UE with that of the static ground BS and mobile UE. It was deduced that the aerial system resembles a single-tier terrestrial cellular network with static BSs and mobile UEs. The handover probability of the two cases was then computed. A numerical analysis was used to investigate the proposed solution based on the Monte Carlo simulation. The Monte Carlo simulation is a computational algorithm that relies on repeated random sampling. The simulation outcomes revealed that, for the Same Speed Model (SSM) case, the terrestrial scenario mirrored the static UEs and mobile drone BSs to resemble static BSs and mobile UEs. Based on the graphical representation of the results, it can be seen that the SSM and Different Speed Model (DSM) initially behaved similarly. As time passed, the handover probability of SSM became higher than that of DSM. It should be noted that this does not necessarily imply that the handover rate of SSM is higher than that of DSM.

### 6.2. Machine Learning-Based Technique

Due to the high mobility of drones, it is difficult to accomplish global information exchange due to unnecessary overheads since the network becomes significantly convoluted. With the advancements in machine learning disciplines, the number of applications throughout various fields has also increased. Several previous studies were conducted concerning the applications of ML in drone networks.

In 2020 [32], the authors focused on real terrestrial network data for Stockholm. System models were developed using Key Performance Indicators (KPIs), communication delay, and interference to simulate the Handover and Radio Resource Management (H-RRM) optimization problem. This issue was then converted to a machine learning problem, presenting a reinforcement learning solution that detects the temporal and spatial level connections to produce seamless handover decision The system model was performed by the air-to-ground channel where the LoS pathway is predominant. The buffer line was utilized to identify the upcoming data rate, the specified band, and the interference from the BSs. The optimization problem was then developed, and the algorithms were used to execute the decision-making task. The overall handover process was also updated.

In 2021 [123], the authors have addressed the optimal location of multiple DBSs in a MIMO wireless network setting. They developed a low machine learning-based algorithm for optimizing DBS location by minimizing the total wireless RSS occurring by active terminals. When compared to the Euclidean cost benchmark, the proposed algorithm decreases the propagation loss in the system and achieves a lower bit error rate. Nonetheless, energy-related issues have not been fully covered.

### 6.3. Deep Learning-Based Technique

Deep learning is currently becoming a key solution technology for addressing connection and mobility challenges for connected drones. Moreover, deep reinforcement learning utilizes the combination of deep learning techniques and reinforcement learning principles. Most current research focuses on deep learning/machine learning-based techniques. With the help of developments in the AI field, handover decision making, and other features (such as security challenges), these techniques can now ensure further enhancements [37]. Since learning user behavior does not require periodic updates, the predictions’ precision and efficiency of resource allocation can be improved. Recent years have witnessed overwhelming developments in deep learning methods. It is now possible to integrate the underlying methods in drone networks. Accordingly, various studies have been conducted to address different issues; some of these are summarized below.

In 2017 [33], the authors conducted research on the handover mechanism based on deep learning for UAV networks. The aim of the work was to obtain seamless handover by utilizing deep learning mechanisms. The model’s operation is based on trajectory predictions. The drone altitude was included by measuring the handover rate in 3D and LTE BSs were used. The assessment method consists of three fundamental steps, as suggested in [124]: handover establishment, handover employment, and handover culmination. The BS handoff was determined according to the TTT criterion using a reference power level. The position of the drones was defined using the trajectory prediction algorithm. A threshold function was then utilized to decide whether or not the BSs should switch between drones. The trajectory prediction method was implemented using neural networks instead of the Gaussian regression method, which is generally employed. For the system to have memory, the recurrent neural network (RNN) model and a supervised learning algorithm were applied. The location of the new data was determined by minimizing the root mean square error (RMSE) of the distance between the output generated by the method and the target output values obtained from the training data pairs. As previously mentioned, a threshold function was used based on the TTT values to determine whether or not handover should be performed. As with conventional handover mechanisms for ground UEs, the system’s efficiency decreases as the TTT value increases. The RNN model proved to be more efficient than the traditional handover mechanisms, achieving a higher success rate and lower overhead assessment in overlapping regions. Table 3 also summarizes the most recent works on drone connectivity-related issues.

In 2018 [87], the authors conducted research on deep reinforcement learning for user access control in UAV networks. The authors proposed deep reinforcement learning as a framework solution to address the access control challenges for the ground users with the consideration of mobility of UAV-BS. They aimed to enable the user to be able to intelligently perform access decisions, and maximize users’ data rate and reduce the handover rate as much as possible. Based on the presented simulation results the authors reported that the results illustrated the effectiveness of the proposed solution and exposed its gain as compared to the other selected benchmark solutions from the literature.

In 2019 [63], the authors conducted research on multi-user access control in UAV networks based on deep learning technology. In this work, a deep reinforcement learning method was proposed to provide a centralized control of multiple users in order to enhance the data rate and avoid unnecessary handovers. The proposed method was based on the concept of enabling each user to make its independent access decisions depending on the network information. Frequent handovers must be avoided without compromising the operation of the UEs. The sole purpose is to maximize the throughput amount. The algorithm was implemented using the Markov decision process. The simulations revealed that, by utilizing deep reinforcement learning, the throughput is maximized by employing fewer handovers than the three other commonly used methods, i.e., the RSS-based technique, Q-learning, and Unit Control Block (UCB) learning.

In 2020 [125], the authors provided a novel DQL model for optimal deployment of a UAV-BS. Furthermore, the proposed method presents the optimal UAV-BS trajectory while ground users move without re-learning the method or acquiring ground user path information. In particular, the model optimizes the trajectory of a UAV-BS by achieving a maximum Mean Opinion Score (MOS) for ground users who move along various paths.

In 2021 [126], the authors conducted research on a deep learning technique for drone networks based on THz bands considering handover and beam prediction. The work target was to investigate the utilization of THz bands for drone networks as this band can achieve high data rates, which is a main requirement in future mobile networks. However, the implementation of THz bands faces various technical issues, such as the high path loss, channel impairments, and blockage affect, which become more critical issues when counting the mobility challenges of drones. The authors proposed a deep learning technique that proactively forecasts the candidate-serving base station/RIS and the candidate-serving beam for each connected drone. The model makes the prediction based on the previous observations of drone locations/beam trajectories. The proposed technique relies on a recurrent neural network, which is known as the Gated Recurrent Unit (GRU). The work utilizes the reconfigurable intelligent surfaces (RISs) for addressing the challenges of handover and beam selection when the terahertz (THz) frequency is utilized for drone communication networks. They integrate RISs into THz drone communications because the RISs offer flexibility to extend communication coverage by adjusting to channel dynamics. This proposed solution is able to enhance the drone’s coverage further, which leads to improving the communications reliability of upcoming mobile technologies. Forecasting the next candidate target beams depending on the drone beam/location trajectory contributes to significantly decreasing the beam training overhead and its related latency, and thus appears as a practical solution to serve time-critical use cases. Based on the presented simulation results, the authors reported that the proposed deep learning technique is a promising solution for future RIS-assisted THz networks by reaching near-best proactive handover performance with accuracy exceeding 90% for beam prediction.

In 2022 [127], the authors conducted research on handover decisions based on deep reinforcement learning technology for UAV networks. This paper presents a new handover decision technique based on deep reinforcement learning to avoid the occurrence of unnecessary handovers as much as possible while upholding reliable and stable communication. The proposed solution takes the UAV state as an input for a proximal policy optimization technique and develops a Received Signal Strength Indicator (RSSI) based on a reward function for the online learning of UAV handover decisions. The proposed technique was evaluated with various system settings and mobility scenarios, and with the consideration of UAV movement in 3D. Based on the simulation results and the reported discussion, the proposed technique provides significant enhancements by decreasing the unnecessary handovers by up to 73% and 76% as compared to Q-learning and greedy handover decision techniques, respectively. Moreover, the proposed technique ensures reliable and stable communication with the UAV by maintaining the RSSI above −75 dBm more than 80% of the time.

### 6.4. Other Related Studies

In 2018 [128], the authors conducted a survey study focused on “On-Demand” architecture for an Ultra-Dense Cloud-Drone Network (UDCDN) architecture. This architecture can meet the needs of the next generation by addressing interference issues, energy consumption limitations, front and backhauling challenges, etc. System optimization for issues such as interference, efficiency, and HO performance can be accomplished by integrating UDCDN with terrestrial networks operating in the sub-6 GHz and mm-wave bands, which are not covered in this study.

In 2019 [129], the authors studied the trajectory strategy and power control for multi-UAV networks. They considered machine learning technology as a key approach. This study proposed a three-step method based on machine learning techniques to obtain both the position information of users and the trajectory design of the UAV. However, the UAVs’ energy is limited, and handover between UAVs was not discussed.

In 2019 [130], the authors conducted study on interference modeling for UAV networks depending on stochastic geometry. They developed stochastic geometry-based models. The authors developed models of drone cellular networks based on stochastic geometry. The result shows that the interference becomes more homogeneous as time approaches infinity. The rate decreases as one’s height increases.

In 2019 [131], the authors conducted research on adapting a modulation and coding scheme based on deep reinforcement learning in cognitive mobile heterogeneous networks. The authors addressed the issue of global network information exchanges in drone systems. Similar to the previously discussed paper in [127], a deep reinforcement learning method was designed to manage the UE’s access decisions and maximize network throughput without compromising the handover performance process. The user access model relies on the recipient power. The channel was designed with consideration of the LoS and NLoS components. Deep reinforcement learning techniques were employed for two fundamental features: the training data can be obtained by constant observation of the environment, and long-term benefits are acquired rather than instantaneous ones [132].

In 2021 [133], the authors studied the tracking of autonomous UAVs in surveillance use cases. They proposed drone integration in different areas. These studies proposed integrating drones into the lives of people with special needs, and in telepresence, surveillance, and delivery systems. The proposed systems also have drawbacks, such as being susceptible to weather conditions. Furthermore, these studies lack simulation modeling and results that would aid in comparing this concept to other existing solutions.

In 2022 [134], the authors studied the power-efficient wireless coverage using the minimum number of UAVs. They proposed wireless coverage with the minimum number of UAVs using less power. This article proposed a method involving multi-UAV 3D deployment with power-efficient planning, which was introduced with the goal of reducing the number of UAVs used to provide wireless connectivity to all outdoor and indoor users while significantly reducing the required UAV transmit power and meeting users’ data rate requirements. This involved high computational complexity, which was manifested in terms of the algorithm execution time. Moreover, handover between UAVs was missing

In 2022 [135], the authors mainly focused on fast multi-UAV path planning for optimal area coverage in aerial sensing applications. They proposed fast Coverage Path Planning (CPP) for multiple UAVs. A software framework and an algorithm were used to solve and analyze the problem. According to the results, the method obtains optimal UAV paths to complete the overall mission within the minimum time. The study only tested routes generated in 2D space at a constant altitude. As a result, there is a higher probability of handover, in addition to other mobility-related issues.

**Table 3 sensors-22-06424-t003:** A summary list of related works on drone mobility management and connectivity.

Ref	Year	Study Focus	Proposed Method	Solution Target	Environment
[59]	2019	Experimental work on handover	Performance evaluation based on experimental data	Study the effect of cell selection on handover	LTE-A network
[106]	2012	Mobility	Performance evaluation	Seamless horizontal and vertical mobility	VANET
[107]	2011	Mobility/handoff	Survey study	State of the art on mobility	Vehicular networks
[108]	2015	Coverage and handover control	Algorithm based on RSS, regulates the coverage of each drone.	Optimal coverage control and efficient handover	Drone networks
[109]	2019	Handover	Survey study	State of the art on handover	Vehicular ad hoc in 5G mobile networks
[110]	2016	Handover	Handover scheme to adjusts the height of a drone and the distance between the drones.	Handover management	Drone networks
[111]	2016	Cell-selection optimization handover	A multiple-criteria decision-making based on an integrated fuzzy technique	Cell-selection optimization handover	Long-Term Evolution (LTE)
[112]	2017	Handover optimization	Self-optimizing algorithm for handover detection, execution and decision parameter	Self-organizing method for handover performance optimization	LTE-Advanced network
[113]	2017	Fuzzy interference system	Fuzzy inference	Intelligent handover scheme	Drone network
[114]	2018	Classification of movements for mobility Prediction	This paper proposed a machine-learning-based solution for classifying mobility based on predicted node locations in the near future.	Mobility prediction and object profiling	Drones in UAV networks.
[115]	2019	Handover Probability	Tractable equivalent model and handover probability through stochastic geometry analysis	Equivalent model for 3D UAV networks.	UAV networks
[116]	2019	Mobility	Performance analysis based on stochastic geometry	Analysis under random waypoint mobility model	Drone cellular network
[117]	2019	Mobility Model for a Drone	Performance evaluation based on stochastic geometry	Mobility analysis	3GPP-drone cellular network
[118]	2019	Mobility Support	Performance analysis	Experimental work for mobility	Cellular connected UAVs
[119]	2019	Route-aware handover enhancement	Algorithm based on path information	- Optimize the network using flight path data. - Reducing HOF	Drones in cellular networks
[120]	2019	Optimization of packet routing	Algorithm based on priority, time to live, and power consumption constraints	Novel DTN mobility algorithm improves packet driven routing	Autonomous drone logistics networks
[121]	2019	Mobility in mm-wave/THz bands	Performance analysis	Effects of mobility uncertainties on mm-wave/THz band	Drones in the mm-wave/THz bands
[122]		Location module for tracking to support mobility management of drones	A location module that can be integrated in Sensor Gateways and 5G BS	Location module to monitor UAVs and learn about their state while they are moving	Drones in 5G networks
[123]	2021	Location strategy for Drone base stations	Machine learning and performance analysis	Address the optimal positioning of multiple DBSs	Heterogeneous networks
[125]	2021	UAV trajectory design considering mobile ground users	Deep Q-network (DQN)-based learning	Optimizes the trajectory of a UAV-BS by maximizing the mean opinion score (MOS) for ground users	5G networks
[126]	2021	Beam and handoff prediction	Deep learning solution based on a recurrent neural network, namely the Gated Recurrent Unit (GRU)	Extend the coverage of drones and enhance the reliability of next-generation wireless	Terahertz (THz) drone networks
[127]	2022	Handover decision	Deep reinforcement learning	Avoid unnecessary handovers upholding reliable and stable communication	UAV networks
[128]	2018	On-demand on Ultra-Dense Cloud Drone Networks	Survey	Presented an Ultra-Dense Cloud-Drone Network (UDCDN) architecture	Ultra-Dense cloud Drone Networks
[129]	2019	Trajectory design and power control for UAV	Machine learning	Obtain the position information of users and the trajectory design of UAV.	UAV-Wireless Networks
[130]	2019	Interference modeling for UAV networks	Stochastic geometry	Efficient interference modeling	Drone Cellular Networks
[131]	2019	Modulation and coding scheme selection	Deep reinforcement learning	Efficient selection for modulation and coding scheme	Cognitive Heterogeneous Networks
[132]	2021	Mobility in drone taxi applications	Deep reinforcement learning	Compute the optimal transportation routes	UAV mobile network
[133]	2021	Dynamic object tracking on UAV system	A learning-based UAV system	Achieving autonomous surveillance	UAV mobile network
[134]	2022	Power-Efficient Wireless Coverage of UAVs	Multi-UAV 3D deployment with power-efficient planning	- Reducing the number of UAVs used to provide wireless connectivity - Reducing the transmit power - Meeting users’ data rate requirements.	UAV mobile networks
[135]	2022	Fast Multi-UAV Path Planning for Optimal Area Coverage	Software framework and an algorithm	Obtains optimal UAV paths to Complete the overall mission at the minimum time.	UAV mobile networks
[136]	2018	Drone-delivery using autonomous mobility	Drone-delivery using autonomous mobility (DDAM)	Solve: (1) high demand of delivery; (2) short delivery lead-time; and (3) complex traffic congestion.	-
[137]	2020	Performance characterization of mobility models	Performance analysis	Characterize the performance of several canonical mobility models in a drone cellular network	Drone cellular Networks
[138]	2020	Mobility and service-oriented modeling	Neuro-fuzzy interference system	Assist in reliable and efficient route selection	Ad hoc networks
[139]	2021	Optimization for drone mobility	Q-learning	Optimize handover decision regularly to provide efficient mobility support with high data rate in time-sensitive applications, tactile Internet, and haptics communication	5G and Beyond Ultra-Dense Networks
[140]	2020	Drone mobility support	Reinforcement learning/Q-learning algorithm	Ensure robust wireless connectivity and mobility support for drones in the sky	Long-term Evolution (LTE) and the Fifth-Generation New Radio (5G NR)

## 7. Future Directions

Drones are not yet widely available. It will take time to fully integrate connected drones into serving communication networks. A number of potential recommendations and major research directions should be addressed before the wide employment of connected drones to mobile networks. Accordingly, this section highlights and discusses a number of key research directions related to mobility management of drones over mobile networks. These key research directions must be addressed efficiently to enable more efficient connected drone service over wireless networks.

### 7.1. Energy Efficiency

As previously mentioned, one of the significant challenges facing drone networks is the limited power, which may lead to the termination of drone operations in certain cases. Limited power is a substantial challenge that may also lead to increasing the frequency of handovers. Connected drones require more power as compared to terrestrial UEs due to their connections and movement characterizations. For example, when drones move in 3D with high speed, the handover probability will increase. This leads to increasing the handover signaling, which in turn leads to more power consumption. Therefore, to reduce the power consumption of drones, more efficient mobility techniques, and energy-efficient techniques must be used. Renewable energy methods, which are also effective solutions, should be considered, especially for those remotely controlled from long distance. This is a research direction that can be pursued in the future since more effective solutions are required.

### 7.2. Mobility Management

In upcoming HetNets, mobility control of drones will be a critical aspect that requires thorough analysis. A major risk exists since drones rapidly move in three dimensions with high speed and different characterizations [114,137,138]. This increases the handover probability and may lead to increasing the handover ping-pong effect and RLFs. Another significant issue during drone movement is the use of the mm-wave spectrum and terahertz band; this use in next-generation networks is discussed in [141]. The rapid development and massive growth of drones and mobile networks will further exacerbate the problem since load balancing will be a critical factor, necessitating an appropriate solution. The case becomes more critical if no optimal and efficient handover mechanisms are used. Therefore, managing the connection during drone mobility must be adequately emphasized in future networks.

### 7.3. Machine Learning for Drones

Machine learning technology is a key technique that provides efficient solutions in wireless networks. The capability of this technology can be a key solution for mobility management issues of drones but deeper investigations are required. By offering training, this method provides continuous learning and improvement. The understanding of the drone’s environmental impact will be enhanced with further research. This will enable unmanned aircraft systems to improve even further. Thus, this is a potential pathway to enable drones to become key connected components in future mobile networks.

As an example, machine learning has been examined as a suitable technology for mobility prediction of drones, as investigated in [129]. Some research has been conducted in the literature to investigate the efficiency of machine learning for addressing mobility issues of connected drone networks [139,142,143]. However, to the best of our knowledge, sufficiently deep and numerous research works have yet to be conducted that can be considered to be comprehensive and efficient solutions for addressing the existing challenges. Thus, the work in this direction will be a key research area that needs to be examined in future research.

### 7.4. Deep Learning for Drones

Similarly, deep learning technology is a promising solution that can be used to address mobility management issues of drones in mobile networks. Research has also been conducted in the literature to investigate the efficiency of deep learning for addressing the mobility issue of connected drone networks [140,142]. Further enhancements with the use of these techniques can be achieved with the latest developments in the AI field, handover optimization, handover load balancing, handover decision making, and other aspects [37]. The precision and efficiency of predictions for resource allocation can be improved. Recent years have witnessed an overwhelming development of deep learning methods. It is now possible to integrate these fundamental methods in drone networks to address motility issues.

### 7.5. IoT and Drones

Since IoT and drones can support low-cost platforms and services, their combination will certainly be a future component [84]. The increasing growth in IoT and the great need for higher data rates and low latency will likely necessitate the use of drones. Drones can contribute to significant solutions in several IoT use cases. Future networks should therefore incorporate the latest research and enhancements. However, the massive increase in these technologies will also increase the issues related to mobility. Higher handover probabilities may occur. Moreover, the need to balance load between the serving cells will increase. Thus, the implementation of drones in IoT use cases will need to be investigated further.

### 7.6. MANETs and VANETs Applications in Drone Networks

In future networks, Flying Ad Hoc Networks (FANETs) will be an active technology in mobile networks to enable drones to provide various services over a wide communication range. These drones will need to communicate directly or indirectly depending on the communication range to secure more reliable communications. This can be directly, if the two connected drones are located within a close communication range, or indirectly over a number of drones relay nodes, if they are distant. The concept is similar to that of previous technologies in the fields of MANETs and VANETs. However, setting up FANETs will be more challenging as compared to the traditional networks, such as Mobile Ad hoc Networks (MANETs) and Vehicular Ad hoc Networks (VANETs). The requirements will be different in terms of node mobility, connectivity, message routing, service quality, application areas, and other necessities. Therefore, the introduction of FANETs models, analyzing opportunities, identifying open research issues, and addressing the challenges in FANETs will comprise a key research direction in future mobile networks. Various mobility situations and system settings over various deployment scenarios will be more challenging.

### 7.7. New Cellular Technologies

New challenges have emerged as a result of the latest generation of cellular technology, leading to an increase in network heterogeneity. For example, the implementation of 5G and 6G mobile networks will contribute to an increase in mobility issues. This is because these technologies will mostly operate based on high-frequency bands, which will lead to reducing the cell coverage. This, in turn, will increase the handover probability. This will be even greater in the case of drones, because drones move in three dimensions with high movement speeds and mostly with LoS connections. Thus, the handover probability will definitely increase further. Moreover, future mobile networks will be characterized as ultra-dense heterogeneous networks. Various mobile technologies will be deployed as overlapping with each other’s. This also will increase the handover probability, especially if drones have the capabilities to be connected to more technologies. Thus, effective and more intelligent handoff algorithms must be implemented to resolve these challenges. With the launch of various cellular technologies, drones can be used to enhance 5G spectral efficiency. Although the application of drones in 5G networks is still at its infancy stage, interest in such integration is rapidly growing.

### 7.8. Security

One of the most fundamental issues for any digital system is security. If a drone BS is interrupted by an attacker, for instance, the UEs served by that drone BS are more likely to lose connection than the UEs served by ground BSs. If a drone is operated by attackers, the UEs supplied by terrestrial BSs may face significant interference due to LoS links. When drones are utilized for cellular communications, it is critical to ensure the security of drone systems. The security and safety issues will become more critical with small drones, and the massive growth in drones having fast movements and the capability of long-distance travel. Drones’ security vulnerabilities and threats are still a challenge that need further study. Therefore, drone security and privacy concerns with various mobility scenarios must be highlighted, discussed, and addressed, particularly drone vulnerabilities, threats, and attacks. Therefore, further research and enhancements must be accomplished in this area.

### 7.9. Mobile Edge Computing with Drones

Mobile Edge Computing is a new cellular network scheme in which BSs provide connections to UEs and computing services. This technique essentially brings cloud services closer to UEs, reducing latency for several compute-dense applications such as speech recognition and augmented reality. When MEC is supported by drone BSs, a number of issues arise. Drone BSs must have certain computing platforms, such as Graphical Processing Units (GPUs), to provide cloud services that will improve drone energy consumption and payload. Another issue is computing session continuity since fast-moving drone BSs may create serious disconnections for the ongoing computational functions of UEs. Further research should be conducted to address the potential of MEC and its challenges regarding drones [143]. Different system settings with various mobility scenarios should be considered.

### 7.10. Drone Antennas

Drones can travel in three dimensions at various speeds. There is an urgent need to develop a new tracking antenna that can adapt to drone mobility and enable an ultra-high data rate transfer between drones and BSs. Since the accelerometer, gyro, and GPS data are used to track BSs, the antenna is tilted [144]. Another challenge is the limited area available for antennae on drones, particularly for small drones. Additional research and development must be prioritized in this regard.

## 8. Conclusions

This paper mostly focused on studying handover managements in drone networks. Drones are a popular alternative to ground-based BSs or UEs. Due to limited power consumption, packet loss, or dense networks, various challenges may emerge during drone operation, making the handover process critical for effective data transfer. A comprehensive review of previous research was presented and discussed. Various research challenges were also highlighted. The proposed solutions from the literature were extensively reviewed. The main focus, however, is on handover management in future mobile networks. From this overview, several points can be highlighted. The research trends indicate that in future mobile networks, the integration of drones in mobile cellular networks, satellite networks, and other traditional technologies (such as MANETS, VANETs, and IEEE 802.11) will be part of the main solution. Drones move at higher speeds than ground network UEs and have different characterizations. Initially, it would seem that the two network behaviors have several similarities; however, drones possess greater handoff probability. This will lead to further handover issues, such as high HPPP and RLF. This is a significant challenge facing the implementation of drones. Existing handover mechanisms may not be efficient for drone networks. Machine and deep learning-based handover models have higher success rates and fewer assessment overheads in overlapping regions than traditional approaches, indicating that this technology may be a successful solution for managing the handover issue of drones.

## Figures and Tables

**Figure 1 sensors-22-06424-f001:**
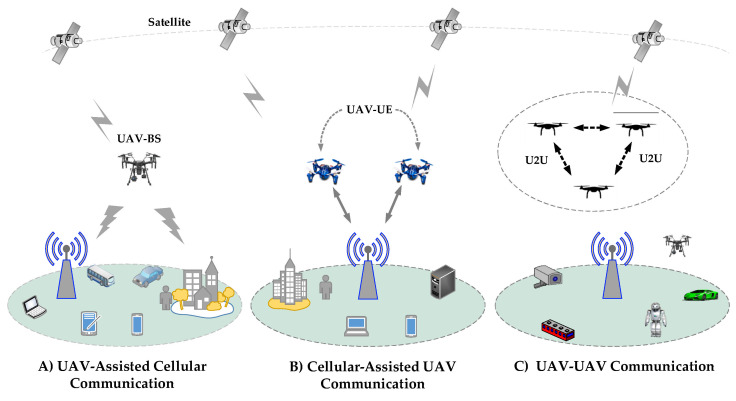
Drone system architecture, solutions, and integration in future mobile networks.

**Figure 2 sensors-22-06424-f002:**
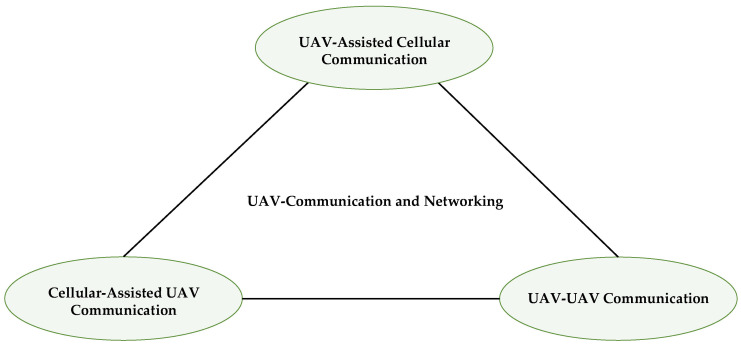
Drones and cellular network integration opportunities.

**Figure 3 sensors-22-06424-f003:**
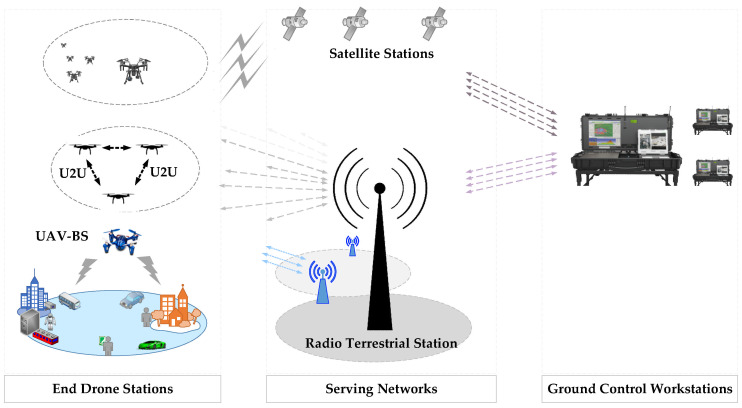
Drone system architecture.

**Figure 4 sensors-22-06424-f004:**
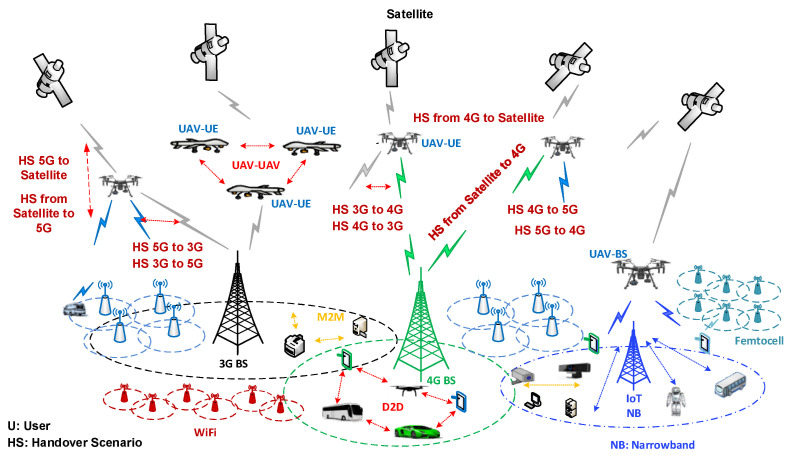
Handover scenarios with connected drones in future mobile networks.

**Figure 5 sensors-22-06424-f005:**
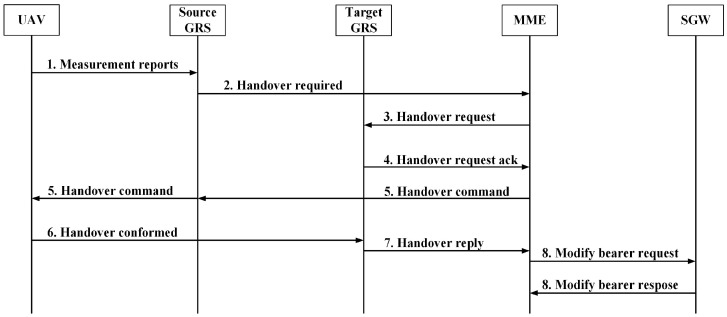
S1 key renewal process in drone networks.

**Figure 6 sensors-22-06424-f006:**
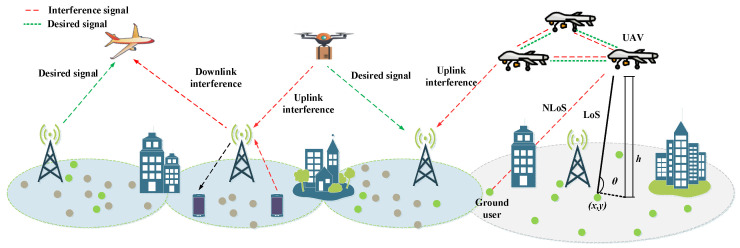
Interference level with connected drones.

**Figure 7 sensors-22-06424-f007:**
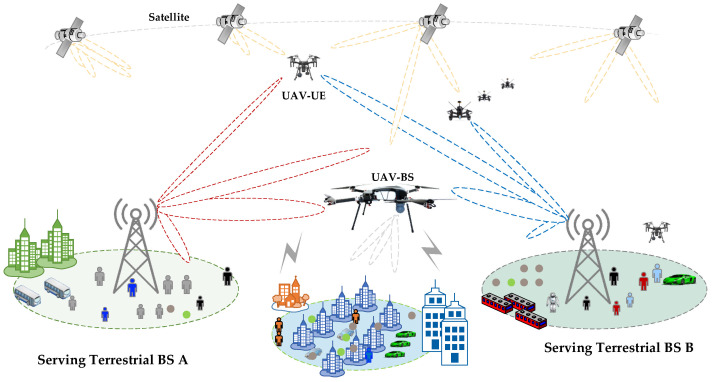
The antenna beam pattern and antenna sidelobes effect.

**Table 1 sensors-22-06424-t001:** List of abbreviations.

Item	Descriptions	Item	Descriptions
2D	Two Dimensional	OWC	Optical Wireless Communication
3D	Three-Dimensional	RRC	Real-Time Control
3G	Third Generation	mm-wave	Millimeter Wave
4G	Fourth Generation	MR	Mixed Reality
5G	Fifth Generation	m-Wave	Micrometer-Wave
6G	Sixth Generation	NCHO	Network-Controlled Handoff
ABSs	Aerial Base Stations	NEMO	Network Mobility
AI	Artificial Intelligence	NLoS	Non-Line-Of-Sight
AMF	Access And Mobility Management	PCI	Physical Cell Identity
API	Application Programming Interface	PGW	Packet Data Network Gateway
AR	Augmented Reality	PMIP	Proxy Mobile IP
AuC	Authentication Center	PPP	Poisson Point Process
BSs	Base Stations	PPs	Ping-Pongs
CoA	Centroid Of Area	QoS	Quality Of Service
CoMP	Coordinated Multi-Point	REHO	Reduced Early
D2D	Device-to-Device	RL	Reinforcement-Learning
DBS	Drone Base Stations	RLFs	Radio Link Failures
DRL	Deep Reinforcement Learning	RNN	Recurrent Neural Network
DSM	Different Speed Model	RSRP	Reference Signal Received Power
FAA	Federal Aviation Administration	RSS	Received Signal Strength
FMIPv6	Fast Mobile Ipv6	RSSI	Received Signal Strength Indicator
GCS	Ground Control Station	RWP	Random Waypoint
GPS	Global Positioning System	S-BS	Serving Base Station
GPUs	Graphical Processing Units	SDN	Software-Defined Network
HD-SOHP	Handover Detection Self-Organizing Handover Parameters	SGW	Serving Gateway
HetNets	Heterogeneous Networks	SINR	Signal-to-Interference-Plus-Noise Ratio
HMIPV6	Hierarchical Mobile IPv6	SRSs	Sounding Reference Signals
HOF	Handover Failure	SSM	Same Speed Model
HOs	Handovers	STRAW	Street Random Waypoint
HOM	Handover Margin	T-BS	Target Base Station
H-RRM	HO And Radio Resource Management	TCP	Transmission Control Protocol
HSS	Home Subscriber Server	TTT	Time-to-Trigger
IDT	Internet of Drone Things	U2I	UAV-to-Infrastructure
IIoT	Industrial IoT	U2U	UAV-to-UAV
IoE	Internet of Everything	UAVDRONEs	Unmanned Aerial Vehicles
IoT	Internet of Things	UCB	Unit Control Block
KPIs	Key Performance Indicators	UEs	User Equipment
LAANC	Low Altitude Authorization and Notification Capability	UPF	User Plane Function
LAN	Local Area Network	URLLC	Ultrareliable Low-Latency Communication
LoS	Line-of-Sight	UTM	Unmanned Aircraft Systems Traffic Management
LTE-A	Long-Term Evolution	V2V	Vehicle-to-Vehicle
MAHO	Mobile-Assisted Handoff	V2X	Vehicle-to-Everything
MANETs	Mobile Ad Hoc Networks	VANETs	Vehicular Ad Hoc Networks
mcMTC	Mission-Critical Machine-Type Communication	VIP	Vehicular IP
MEC	Mobile Edge Computing	VR	Virtual Reality
MIH	Media Independent HO	Wi-Fi	Wireless Fidelity
MIMO	Multiple-Input Multiple-Output	WLAN	Wireless Local Area Network
MIPv4	Mobile IP Version 4	WMNs	Wireless Mesh Networks
MME	Mobility Management Entity	XR	Extended Reality

**Table 2 sensors-22-06424-t002:** Outcomes of the algorithms.

Algorithms	No. of Handover (Random)	No. of Handover (Straight)
Conventional	13.86	5.03
Work done by [113]	0.84	2.37

## Data Availability

Not applicable.

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
