# Peer review of "Handover Management for Drones in Future Mobile Networks—A Survey"

_sensors, 2022, doi:10.3390/s22176424_

Round 1

Reviewer 1 Report

In the reviewer opinion the paper is now more coherent. Methodologies, structure and organization are clearer. In particular, the abstract and introduction significantly clarify the main contributions of the paper. The organization is good and a good future direction section is provided.

The paper is clear and it represents a good review on this specific topic

Author Response

Dear Reviewer so many thanks for the Reviewer opinion.

Although there is no action is required, we have made some additional enhancements in the paper. mostly related work section been improved totally and mostly all figures (Except Figure 5), please see the updated manuscript.

 Kind regards 

Reviewer 2 Report

Overall : Accept after minor revision

This paper deals with a variety of scopes in Handover for Drone network. As it is, it's quite well organized and described clearly. However, I suggest some comments on this paper. 

1. Not clear sentence found in Abstract. Need check the typos.
eg. So, mobility issues and handover process among.... (sentence is not clear). At page 8, UAV-TO-UAV (U2U) should be revised as UAV-to-UAV like UAV-to-Infrastrucuture (U2I). 

2.  Figure 2, Figure 3 and Figure 4 are not clear, a bit crude. Font and size should be chosen with more consideration. 

3. At Subsection 5.3 interference Probability, when authors don't mention on detailed description on right figure of Figure 6 (angle notation). Also authors don't mention the references.

4. Subsection 6.2 Machine Learning/ Deep Learning-Based Technique: Authors should separate 'Machine Learning' and 'Deep Learning' Techniques subsection for clear understanding. 

5. Table 3 is very important contribution of this paper. However, it is too wordy, so it is not good to be understandable. It should be summarized based on Meaningful Technique/Network or related Drone Network/ ect. with concise summary.

Author Response

Dear Reviewer 
Best Greeting
So many thanks for your opinions and comments.

We have addressed all the given comments sereiously and we have made some additional enhancements in the paper. We have mostly improved the related work section totally and mostly all figures (Except Figure 5), and table. Please see the updated manuscript.

 Kind regards 

Round 2

Reviewer 2 Report

The authors revised their paper following reviewer's comments. Thus, I suggest this paper is now well qualified to be published after English spell check.

This manuscript is a resubmission of an earlier submission. The following is a list of the peer review reports and author responses from that submission.

Round 1

Reviewer 1 Report

This survey presents the most important information related to handover decision related to Drones. 

In the opinion of this reviewer, the topic is very important. Due to that, it is necessary to include practical applications or cases of study.

Author Response

Reviewer#1

Recommendation:  

Many thanks for the Reviewer opinion, we have addressed all his comments carefully and seriously as illustrated in the following, one by one:

Comments:

Comments and Suggestions for Authors

Reviewer#1, Concern # 1: This survey presents the most important information related to handover decision related to Drones. In the opinion of this reviewer, the topic is very important. Due to that, it is necessary to include practical applications or cases of study.

Author response:  Many thanks for this positive comment.

Author action: Yes, in this manuscript, we focus on handover decisions overview related to drones. Hence, including practical applications or cases of study is beyond the scope of this article. However, we have updated the manuscript briefly with practical applications and case studies in introduction, as well as references that have extensively discussed the practical applications of UAV.

Reviewer 2 Report

The authors discussed different research on handover management for connecting drones to mobile communication networks which is very interesting. However, I have the following concerns.

1. Please grammatical issues of this paper. 
2. Recent UAV-related papers are missing. Some are mentioned as follows. 
-> "A Blockchain-Based Artificial Intelligence-Empowered Contagious Pandemic Situation Supervision Scheme Using Internet of Drone Things," in IEEE Wireless Communications, vol. 28, no. 4, pp. 166-173, August 2021, doi: 10.1109/MWC.001.2000429.
->"FBI: A Federated Learning-Based Blockchain-Embedded Data Accumulation Scheme Using Drones for Internet of Things," in IEEE Wireless Communications Letters, doi: 10.1109/LWC.2022.3151873.
3. Please improve the quality of Fig. 1. 
4. List abbr should be top.
5. Please highlight the novelty of the paper based on the existing works. 
6. Please revised the contribution in bullet form. 
7. Paper should highlight more recent wireless networks, e.g., B5G, 6G, etc.
8. It would be better if authors can discuss security challenges during handover along with privacy issues.
9. Authors should add limitations in Table 2.

Author Response

Reviewer#2

Recommendation:  

Many thanks for the Reviewer opinion, we have addressed all his comments carefully and seriously as illustrated in the following, one by one:

Comments:

The authors discussed different research on handover management for connecting drones to mobile communication networks which is very interesting. However, I have the following concerns.

Many thanks for the Reviewer opinion, we have addressed all his comments carefully and seriously as illustrated in the following, one by one:

Reviewer#2, Concern # 1: 1. Please grammatical issues of this paper.                 

Author response:  Many thanks, we have considered this comment seriously.

Author action: We updated the manuscript by checking the grammar again and sent the paper for proofreading, the Proofread Letter is attached.

Reviewer#2, Concern # 2: 2. Recent UAV-related papers are missing. Some are mentioned as follows. 
-> "A Blockchain-Based Artificial Intelligence-Empowered Contagious Pandemic Situation Supervision Scheme Using Internet of Drone Things," in IEEE Wireless Communications, vol. 28, no. 4, pp. 166-173, August 2021, doi: 10.1109/MWC.001.2000429.
->"FBI: A Federated Learning-Based Blockchain-Embedded Data Accumulation Scheme Using Drones for Internet of Things," in IEEE Wireless Communications Letters, doi: 10.1109/LWC.2022.3151873.                       

Author response:  Many thanks, we have considered this comment seriously.

Author action: We updated the manuscript by considering recent UAV-related papers.

Reviewer#2, Concern # 3: 3. Please improve the quality of Fig. 1.                        

Author response:  Many thanks, we have considered this comment seriously.

Author action: We updated the manuscript by improving the quality of Fig. 1.

Reviewer#2, Concern # 4: 4. List abbreviations should be top.                              

Author response:  Many thanks, we have considered this comment seriously.

Author action: We updated the manuscript by moving the list of abbreviations to the top of the manuscript.

Reviewer#2, Concern # 5: 5. Please highlight the novelty of the paper based on the existing works.             

Author response:  Many thanks, we have considered this comment seriously.

Author action: We updated the manuscript by highlight the novelty of the paper based on the existing works.

Reviewer#2, Concern # 6:  6. Please revised the contribution in bullet form.                      

Author response:  Many thanks, we have considered this comment seriously.

Author action: We updated the manuscript by summarizing our contributions at the end of the introduction (bullet form)

Reviewer#2, Concern # 7: 7. Paper should highlight more recent wireless networks, e.g., B5G, 6G, etc.     

Author response:  Many thanks, we have considered this comment seriously.

Author action: We updated the manuscript by recent wireless networks, e.g., B5G, 6G in section 2.

Reviewer#2, Concern # 8: 8. It would be better if authors can discuss security challenges during handover along with privacy issues.                    

Author response:  Many thanks, we have considered this comment seriously.

Author action: We updated the manuscript by adding a brief related part.

 Reviewer#2, Concern # 9: 9. Authors should add limitations in Table 2 (A list of related works on drone mobility).                

Author response:  Many thanks, we have considered this comment seriously.

Author action: We updated the manuscript by adding limitations to table 2.

Reviewer 3 Report

The paper topic is interesting. The handover issues in UAV connected networks is a key aspect to be taken into account in this kind of network.

In the reviewer opinion the paper is not well written and also the organization can be improved in order to make it more attractive for readers.

First of all, the title is “Handover Decision Algorithms of Drones in 6G Technology and Future Wireless Networks- A Survey”. The authors mentioned 6G technology but in the text no 6G technology is reviewed. Moreover, it is not clear what should be the Future Wireless Network. May be, it should be opportune to change title or introduce in the work the new concepts.

The section “Handover management for drone networks” is very concise, more details could be given by authors.

In section “Handover challenges in drone networks” the subsection RLF, HOFs, Handover decision algorithm can provide more details for a better understanding.

The section Related Work in the current form is just a summary of different works. I suggest to reorganize and provide to give an original form to the section.

Author Response

 Reviewer#3

Recommendation and Comments:

The paper topic is interesting. The handover issues in UAV connected networks is a key aspect to be taken into account in this kind of network.

Author response:  Many thanks for the Reviewer opinion, we have addressed all his comments carefully and seriously as illustrated in the following, one by one:

Reviewer#3, Concern # 1: In the reviewer opinion the paper is not well written and also the organization can be improved in order to make it more attractive for readers.            

Author response:  Many thanks for the reviewer opinion, we have addressed his comment carefully.

Author action: We updated the manuscript by improving the writing and also the organization further.

Reviewer#3, Concern # 2: First of all, the title is “Handover Decision Algorithms of Drones in 6G Technology and Future Wireless Networks- A Survey”. The authors mentioned 6G technology but in the text no 6G technology is reviewed. Moreover, it is not clear what should be the Future Wireless Network. May be, it should be opportune to change title or introduce in the work the new concepts.

Author response:  Many thanks for the reviewer opinion, we have addressed his comment carefully.

Author action: We updated the manuscript by introducing and reviewing a new subsection (section 2) related to drones in future networks. We have also discussed 6G briefly throughout the paper and cited related papers too. We have also updated the title.

Reviewer#3, Concern # 3: The section “Handover management for drone networks” is very concise, more details could be given by authors.

Author response:  Many thanks for the reviewer opinion, we have addressed his comment carefully.

Author action: We have also made some updated related to this part throughout the paper.

Reviewer#3, Concern # 4: In section “Handover challenges in drone networks” the subsection RLF, HOFs, Handover decision algorithm can provide more details for a better understanding.

Author response:  Many thanks for the reviewer opinion, we have addressed his comment carefully.

Author action: We updated the manuscript by regarding this and also, we have cited our related papers to this matter as they have been explained in more details in our other papers.

Reviewer#3, Concern # 5: The section Related Work in the current form is just a summary of different works. I suggest to reorganize and provide to give an original form to the section.

Author response:  Many thanks for the reviewer opinion, we have addressed his comment carefully.

Author action: We updated the manuscript by reviewing the related work section again and made some update in this section.

Round 2

Reviewer 2 Report

I have no further comment. I am recommending to accept this paper. 

Author Response

Reviewer#2

Recommendation:  Comments and Suggestions for Authors

I have no further comment. I am recommending to accept this paper.

Many thanks for the Reviewer opinion.

Author Response: No action is required.

But we have made further enhancements, please see the updated manuscript

Reviewer 3 Report

I do not understand the section "future directions", what is its contribute with the topic of the paper that concerns the handover management??

Title, it is related to 6G but may be it could be deleted and the title could be

"Handover Management Techniques for Drone in Mobile Networks: A survey"

Author Response

Reviewer#3

Comments and Suggestions for Authors

Many thanks for the Reviewer opinion, we have addressed all his comments carefully and seriously as illustrated in the following, one by one. Also, we did further enhancements to the paper.

Reviewer#3, Concern # 1: I do not understand the section "future directions", what is its contribute with the topic of the paper that concerns the handover management?

 Author response:  Many thanks, we have considered this comment seriously.

 Author action: We updated the manuscript by improving the future directions section and link it further to the paper main focus.

Reviewer#3, Concern # 2: Title, it is related to 6G but maybe it could be deleted and the title could be

Author response:  Many thanks, we have considered this comment seriously.

 Author action: We updated the manuscript as Author suggested by changing the title to be:

"Handover Management Techniques for Drone in Mobile Networks: A survey"

We have also made further enhancements, please see the updated manuscript
